# The impact of biomass burning on upper tropospheric carbon monoxide : A study using MOCAGE global model and IAGOS airborne data.

Martin Cussac[1], Virginie Marécal[1], Valérie Thouret[2], Béatrice Josse[1], and Bastien Sauvage[2]

[1]CNRM, Université de Toulouse, Météo-France, CNRS, Toulouse, France
[2]Laboratoire d'Aérologie, Université de Toulouse, CNRS, UPS, Toulouse, France

**Correspondence:** Martin Cussac (martin.cussac@meteo.fr)

**Abstract.**

In this paper the fate of biomass burning emissions of carbon monoxide is studied with the global chemistry transport model MOCAGE and IAGOS airborne measurements for the year of 2013. The objectives are firstly to improve their representation within the model and secondly to analyse their contribution to carbon monoxide concentrations in the upper troposphere. At
5 first, a new implementation of biomass burning injection is developed for MOCAGE, using the latest products available in GFAS biomass burning inventory on plume altitude and injection height. This method is validated against IAGOS observations of CO made in fire plumes, identified thanks to the SOFT-IO source attribution data. The use of these GFAS products leads to improved MOCAGE skill to simulate fire plumes originating from boreal forest wildfires. It is also shown that this new biomass burning injection method modifies the distribution of carbon monoxide in the free and upper troposphere, mostly at northern
boreal latitudes. Then, MOCAGE performances are evaluated in general in the upper troposphere and lower stratosphere in comparison to the IAGOS observations, and are shown to be very good, with very low bias and good correlations between the model and the observations. Finally, we analyse the contribution of biomass burning to upper tropospheric carbon monoxide concentrations. This is done by comparing simulations where biomass are toggled on and off in different source regions of the world to assess their individual influence. The two regions contributing the most to upper tropospheric CO are found to be the
boreal forests and equatorial Africa, in accordance with the quantities of CO they emit each year and the fact that they undergo fast vertical transport: deep convection in the tropics and pyroconvection at high latitudes. It is also found that biomass burning contributes for more than 11 % on average on the CO concentrations in the upper troposphere, and up to 50 % at high latitudes during the wildfire season.

# 1 Introduction

The upper troposphere-lower stratosphere (UTLS) is an important layer of the atmosphere subject to strong gradients in both its dynamics and its chemical composition. Understanding upper tropospheric air composition is necessary to better quantify exchanges of air masses between the troposphere and the stratosphere (Gettelman et al., 2011). Ozone ($O_3$), mostly abundant in the stratosphere, plays a large role on Earth radiative balance (Xia et al., 2018). Its impact on surface temperature for instance, has been shown to be maximum for ozone in the UTLS (Riese et al., 2012). At this altitude, ozone can originate from stratospheric intrusion, but also be produced from tropospheric precursors such as nitrogen oxides ($NO_x$) and carbon monoxide (CO). Even though processes influencing UT air composition are numerous, here we focus on CO, for the reasons detailed below.

Carbon monoxide is one of the primary species emitted during incomplete combustion. It is emitted from both anthropogenic and natural sources. In the atmosphere it is also a product of methane ($CH_4$) and volatile organic compounds (VOCs) oxidation. Since CO is the main sink of tropospheric hydroxyl radicals (OH), it influences the oxidizing capability of the atmosphere (Lelieveld et al., 2016). CO is also a precursor to a few greenhouse gases, such as carbon dioxide ($CO_2$) and ozone. In the upper troposphere, CO is the main precursor of ozone compared other VOCs to other as their lifetime is generally shorter (often a few hours to a few days). Moreover, because ozone radiative impact relies mostly on its distribution in the UTLS (Riese et al., 2012), it makes CO indirectly influencing the global radiative budget of the Earth.

Given its variety of sources, CO can be transported into the upper troposphere by two different mechanisms : (1) When emission occurs in the PBL (Planetary Boundary Layer), usually from anthropogenic origin or from the least intense fires, CO can be transported in the free troposphere and up to the tropopause due to deep convection and to a lesser extent to long range transport, given its 2 months lifetime. (2) For the most active biomass fires, pyroconvection can result in injecting smoke directly above the PBL, and up to 8-9 km for the most extreme cases (Labonne et al., 2007; Kahn et al., 2008). Unlike long range transport, the time scale of these injections is much shorter, and leads to the largest anomalies of CO detected in the free troposphere (Petetin et al., 2018). Since the occurrence of this phenomenon relies heavily on the fire activity, as well as the atmospheric state above the fire, the computational cost to represent this process explicitly in a global chemistry model is high. Biomass burning sources contribute to an important part of CO emitted in the troposphere, and are estimated to range between 350-600 TgCOyr[-1] (van der Werf et al., 2006), which is comparable to CO emissions from anthropogenic sources, estimated between 450-600 TgCOyr[-1] (Lamarque et al., 2010). For comparison purposes, the global in-situ productions of CO from VOCs and $CH_4$ oxidations are respectively estimated to range between 450-1200 TgCOyr[-1] and 600-1000 TgCOyr[-1] (Stein et al., 2014). As biomass burning emissions can reach rapidly the upper troposphere in the form of plume transported though convection or pyroconvection, they have been studied for their potential to contribute to ozone production at this altitude. Enhancements of ozone amounts have been observed and modelled in biomass burning plumes (Thomas et al., 2013), with production increasing while the plume ages. It can lead to export of ozone as the plume are transported by the general circulation on a hemispheric scale (e.g. Brocchi et al., 2017).

Satellite observations have greatly improved our understanding of biomass burning fluxes, resulting in the development of many fire emissions database such as GFAS (Global Fire Emissions Database, Kaiser et al., 2012), GFED (Global Fire Emissions Database, Giglio et al., 2013) or IS4FIRES (Integrated Monitoring and Modelling System for wildland fires, Sofiev et al., 2009). However, biomass burning emissions are still linked to uncertainties compared to other sources of trace gases such as anthropogenic emissions, for three reasons: (1) Their location and time of occurrence, though tied to seasons, is unpredictable. Their detection relies heavily on the frequency and coverage of satellite observations used to make estimates of fire emissions.(2) The nature and amounts of compounds emitted during a fire depend on the nature of the fuel, which can be inferred from a land cover classification map (Friedl et al., 2002). (3) Depending on the intensity of the fire as well as the atmospheric conditions above the fire, pyroconvection can occur (Damoah et al., 2006; Cunningham and Reeder, 2009). This phenomenon can cause the plume to rise above the PBL, and therefore have an important impact on vertical distribution of emissions and therefore plume transport. In this study, we investigate biomass burning emissions and their impacts on CO distribution in the upper troposphere, through global modelling and in-situ measurements.

Different types of measurements are available to study upper tropospheric carbon monoxide. Satellites (MOPPIT for example) provide global measurements, but have been shown to lack vertical resolution, especially in the UTLS, where gradients of CO concentrations are strong (Deeter et al., 2013). Vertical soundings can provide vertically well-resolved data, but concern mostly $O_3$ and lack global coverage. Lastly, aircraft can also be equipped for air composition measurements. Airborne campaigns have helped understanding processes tied to biomass burning thanks to targeted high quality measurements on a regional scale. During the AMMA (African Monsoon Multidisciplinary Analysis) campaign for example, airborne measurements have been used to correctly trace biomass burning plumes over the Gulf of Guinea (Mari et al., 2008) and more generally to study their interaction with the African Monsoon (Haywood et al., 2008). As a complement, the IAGOS infrastructure (www.iagos.org, Petzold et al. 2015) equips commercial airplanes with air composition sensors, in order to provide regular sampling over several decades. This dataset has been chosen for this study since it offers a large number of UTLS multi-species measurements (including CO and $O_3$), on an almost global scale since 1994 for $O_3$ and 2001 for CO.

A challenge to tackle when using the IAGOS dataset is to be able to discriminate the most probable sources of the sampled air masses. As explained earlier, CO sources are multiple, and transport pathways through the troposphere can be various. This is why the analysis of large datasets require other kinds of additional information, like source apportionment, usually obtainable through a Lagrangian backward transport calculation (Seibert and Frank, 2004). SOFT-IO (Sauvage et al., 2017) is a recently developed tool coupling backward transport calculation and emission inventories to estimate the contribution of recent emissions to CO anomalies identified in the aircraft measurements. Results from this model are available as added value products for the whole IAGOS dataset. As this study focuses primarily on biomass burning and CO, data from SOFT-IO are used for source attribution and identification of biomass burning plumes. Identified fire plumes in the IAGOS database are then used to assess the ability of global numerical simulations performed by the MOCAGE (*MOdélisation de Chimie Atmosphérique à Grande Échelle*, Josse et al., 2004; Guth et al., 2016) Chemistry-Transport Model (CTM) to model these plumes.

Regarding the representation of biomass burning within a CTM, previous studies have shown the importance of taking into account the variability in the altitude at which emissions are injected (Turquety et al., 2007; Leung et al., 2007). It was found that

not only the vertical distribution of emitted trace gases was impacted, but also their long range transport as injection can occur directly in the upper troposphere. Moreover, Fromm et al. (2019) recently carried out a reinterpretation of existing literature on the pathway of wildfires emissions to the UTLS, stating that on multiple occasion studies have wrongly attributed plumes observed in the upper troposphere to transport from traditional cumulonimbus (Cb) instead of pyrocumulonimbus (pyCb).

They concluded that the phenomenon of pyroconvection has probably been overlooked and its impact underestimated in past studies, encouraging the use of reliable information on its occurrence to accurately quantify vertical transport of emissions. Recent developments have been made in fire emissions inventories such as the Global Fire Emissions Database (GFAS). They now include plume rise parameters, like the injection height or the top of observed plumes (Rémy et al., 2017). These plume rise parameters are obtained through satellite measurement of the Fire Radiative Power (FRP) and a plume rise model. These

parameters have very recently been used in two different global chemistry models. In the C-IFS model, plume rise parameters from the GFAS inventories have been implemented on a daily resolution and validated against aerosol extinction coefficient measurements during two different airborne campaigns (Rémy et al., 2017). They also found that these products improved in particular the forecast of high altitude biomass burning plumes (above 4 km of altitude). Another approach was tested in the GEOS-CHEM model (Zhu et al., 2018). Monthly vertical profiles of emissions were computed using a plume rise model as

well as satellite data, and used for biomass burning emissions over the United States of America. The results were validated for various trace gases, including CO and PAN. These two studies show an improvement of modelled atmospheric concentrations of trace gases emitted by fires, and encourage efforts to develop the use of such derived fire products for biomass burning emissions, in particular information on vertical distribution. In this paper is presented an implementation recently released GFAS daily products regarding plume rise in the MOCAGE CTM. A systematic global validation of biomass burning induced

plumes is performed over the year of 2013 thanks to IAGOS airborne data, and SOFT-IO information to select CO anomalies originating from biomass burning. A reference year had to be chosen for computational efficiency when comparing IAGOS measurements to model outputs. 2013 was selected as the most recent year with the most validated IAGOS measurements at the beginning of this study. Since we aim to analyse impact on a global scale over a full year, we do not focus on single fire events. Additionally, the results are used to better understand biomass burning impacts on upper tropospheric CO budget, and

relevant process for CO injection at this altitude.

Our objective is to study the impact of biomass burning emissions on upper tropospheric air composition, with a focus on CO exclusively. This work is based on both the IAGOS database and MOCAGE global CTM, presented respectively in section 2.1.1 and section 2.2.1. In section 2.1.2 are presented the SOFT-IO products, and how they are used in this study to automatically identify biomass burning plumes within the IAGOS dataset. The role of injection height during biomass burning

events is investigated in section 3 based on the latest plume rise parameters (both plume top altitude and injection height), taken from the biomass burning inventories GFAS. In section 4, MOCAGE results are evaluated in the upper troposphere against IAGOS observations for the reference year. Finally, the impact of biomass burning at a global scale is analysed in section 5 through MOCAGE sensitivity experiments.

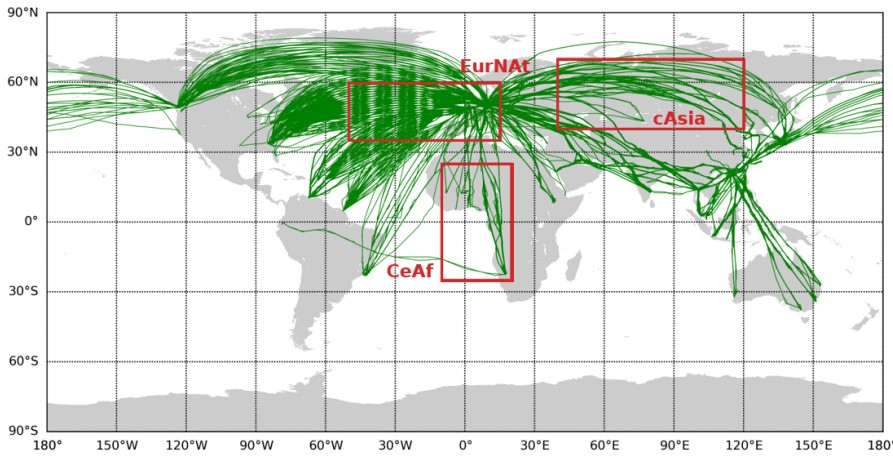

**Figure 1.** Trajectories of the 3022 IAGOS flights that occurred in 2013 (in green), and the delimitations of the three geographical areas used in the evaluation section (in red).

## 2 Methodology

### 2.1 Data description

#### 2.1.1 In-situ measurements : the IAGOS database

IAGOS (In service Aircraft for a Global Observation System) is a European research infrastructure providing a dense database
of in-situ air composition measurements, using medium to long-haul commercial flights. Initially called the MOZAIC program (Marenco et al., 1998), it started in 1994 with aircraft equipped for ozone (Thouret et al., 1998) and water vapour measurements (Helten et al., 1999). Carbon monoxide measurements were begun later in Dec. 2001 (Nédélec et al., 2003). These air composition measurements are complemented with meteorological measurements such as wind direction and speed, and air temperature.

The IAGOS program , (www.iagos.org Petzold et al., 2015) started in 2011 with the same objectives as the MOZAIC program, and the same measurements are performed automatically on board of the equipped aircraft, during the ascent, descent and at cruise altitude. The resulting joint IAGOS-MOZAIC database will be called hereafter the IAGOS database. To complement existing features, greenhouse gases ($CO_2$ and $CH_4$, Filges et al. (2018)) and aerosols measurements (Bundke et al., 2015) are to be instrumented as well in the next few years. On average, 7 to 15 aircraft have been instrumented at the same time to
perform daily flights, and can result in more than 3000 instrumented IAGOS flights per year. Due to the density of the dataset and its timespan, it has been used for different kinds of studies : climatology and trends (e.g. Cohen et al., 2018; Gaudel et al., 2018), chemistry and physics processes (e.g. Ding et al., 2015; Zahn et al., 2014), as well as model evaluation (e.g. Gaudel et al., 2016; Inness et al., 2013; Duncan et al., 2007).

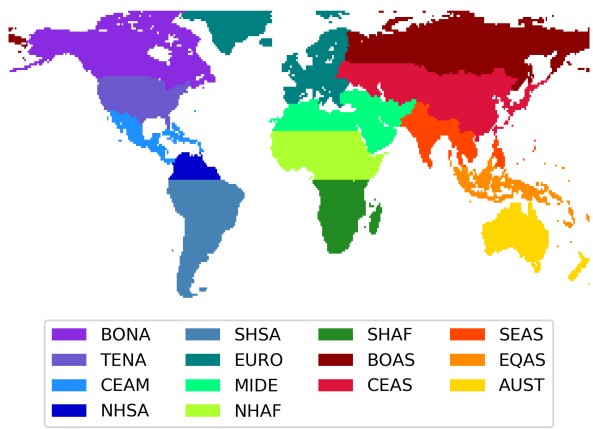

**Figure 2.** Geographical regions defined in the Global Fire Emission Database (GFED), adapted from http://www.globalfiredata.org/data.html

The data used in this study are the CO measurements. Their precision is estimated to be $\pm$ 5 % $\pm$ 5 ppb, for an integration time of 30 s (Nédélec et al., 2003, 2015). Over the year of 2013, a total of 3022 IAGOS flights took place (shown in Fig. 1). It is noticeable that the northern mid-latitudes are the most sampled, with two main axes for IAGOS flights: from Europe to North America going over the Atlantic Ocean, and from Europe to Eastern Asia going over Boreal Asia. The tropics flights tracks mainly cover the African continent and the Maritime Continent. Three geographical areas are also defined for their density in IAGOS measurements, and will be used later in the manuscript for the evaluation of MOCAGE simulations in the UTLS.

### 2.1.2 SOFT-IO products and biomass burning plume identification

SOFT-IO, a tool aiming to identify the most likely source of CO anomalies encountered by the IAGOS equipped aircraft has been recently developed. The complete description of the method can be found in Sauvage et al. (2017). Here is the summary of its main features.

The FLEXPART lagrangian transport and dispersion model is ran along each aircraft trajectory at measurement time and location, for a 20 days backward transport trajectories. Coupled to anthropogenic and biomass burning inventories, it results in an estimation of the contribution of recent emissions (less than 20 days) to the observed CO mixing ratios, but not of the total CO concentrations as it does not simulate CO background. Meteorological fields are taken from the operational ECWMF analysis and forecasts. The calculated CO contribution is decomposed into the contribution from the 14 zones defined in GFED (see Fig. 2). The version of SOFT-IO used in this work is the 1.0.0, where outputs are available using the MACCcity anthropogenic inventory, and for both GFAS and GFED biomass burning inventories. For biomass burning, the contribution computed with GFAS emitted CO amounts and injection height are used in this study, out of consistency with MOCAGE set-up (see section 2.2.3). An example of the SOFT-IO contribution products can be seen in Fig. 3.

SOFT-IO data are used to attribute a source to anomalies encountered by the aircraft. The method used to calculate anomalies is the same as in Sauvage et al. (2017), but only the anomalies measured at cruise altitude are considered, since we focus only

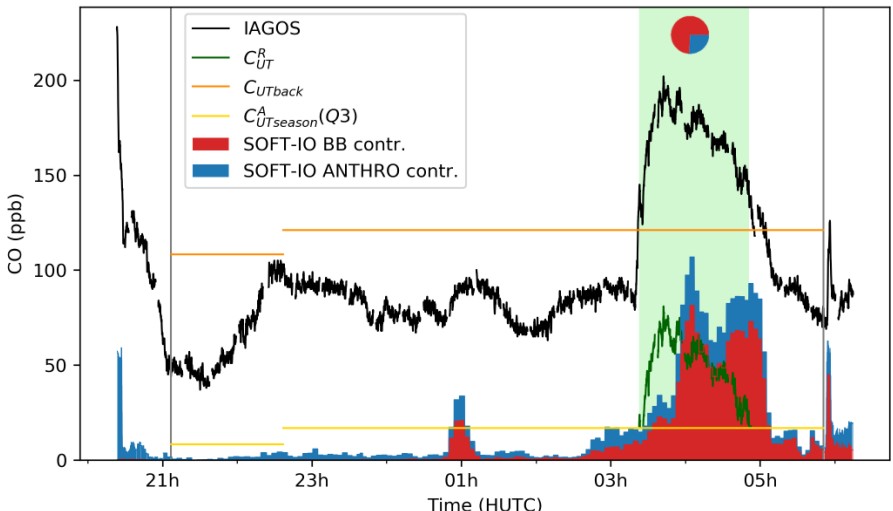

**Figure 3.** Example of SOFT-IO contributions as well as illustration of the method to calculate CO anomalies (see Eq. 1). The selected flight took off from Frankfurt (Germany) the 13$^{th}$ of January 2013 at 20h23 UTC, and landed the next day in Windhoek (Namibia) the next day at 6h14 UTC. The contribution is here at 74% from biomass burning, of which 97% are from NHAF (not shown). The two vertical grey lines represent the end of ascent and the beginning of the descent of the plane. The CO anomaly (in green) is displayed only when superior to the third quartile of anomalies.

on the upper troposphere. CO anomalies $C_{UT}^{A}$ are defined as the difference between the measured CO mixing ratio $C_{UT}$ and the background CO mixing ratio, $C_{UTback}$. $C_{UTback}$ is evaluated as the regional median seasonal value computed for the whole IAGOS database (see Sauvage et al. (2017) for more detail). Finally, only the anomalies superior to the seasonal value of the third quartile of CO anomalies, $C_{UTseason}(Q3)$, are considered. This can be summarized as:

$$C_{UT}^{A} = C_{UT} - C_{UTback} \quad \text{if} \quad C_{UT}^{A} > C_{UTseason}^{A}(Q3) \tag{1}$$

This calculation is only performed when the plane is considered to be below the tropopause, defined here as the 2 PVU (potential vorticity unit) isosurface. In the tropics, where the definition of the potential vorticity diverges, the aircraft is always considered below the tropopause as the highest cruise altitude is under 12 km. The ascent and descent phases of the flight are not considered either. A detailed example of the process is presented in Fig. 3.

Based on both the anomaly calculation and the contribution calculated by SOFT-IO, a method to automatically identify biomass burning related anomalies is used. For each CO anomaly detected during a flight, if the biomass burning contribution from SOFT-IO is on average higher than 5 ppb and is greater than the anthropogenic contribution from SOFT-IO, the anomaly is selected as a biomass burning plume. This ensures that biomass burning provides a significant anomaly with respect to CO background and that this is the main contributor. In total, 220 plumes were sampled by IAGOS aircraft in 2013, with a majority of them originating from the African continent (either SHAF or NHAF) and boreal latitudes (either BOAS and BONA). A summary of the number of plumes by origin can be found in Table 1. The choice was made to merge together

**Table 1.** Number of biomass burning plumes sampled by IAGOS aircrafts in 2013, following SOFT-IO contribution calculations per geographical origin. Regions from which no plume was sampled are not shown. The MULTIPLE origin corresponds to plumes having more than one possible region of origin (other than AFR and BOREAL).

| Origin | Number of plumes |
|---|---|
| AFR (NHAF/SHAF) | 95 (33/62) |
| BOREAL (BOAS/BONA) | 51 (21/30) |
| TENA | 21 |
| CEAM | 14 |
| SHSA | 2 |
| SEAS | 8 |
| MULTIPLE | 29 |
| TOTAL | 220 |

plumes origination from NHAF and SHAF, as well as plumes from BONA and BOAS, as their characteristics are expected to be similar.

## 2.2 Global model

### 2.2.1 MOCAGE global CTM : general description

MOCAGE is the off-line global CTM (Chemistry-Transport Model), developed at Météo-France (Josse et al., 2004; Guth et al., 2016). It is used for a wide variety of research subjects concerning atmospheric air composition, at global and regional scales. It is also operated daily as part of the European ensemble forecasting system in the in the Copernicus Atmosphere Monitoring Service (CAMS), with eight other chemistry models (Marécal et al., 2015).

MOCAGE chemistry is based on 2 different schemes. In the troposphere, the RACM (Regional Atmospheric Chemistry Mechanism) described in Stockwell et al. (1997) is used, while the REPROBUS (REactive Process Ruling the Ozone BUdget in the Stratosphere) scheme is used in the stratosphere (Lefèvre et al., 1994). Merged together they allow for the representation of 112 gaseous species, 434 chemical reactions of which 57 are photolysis reactions. The solver used follows a fully implicit discretization method, described in detail in Cariolle et al. (2017). Aerosols, though described in MOCAGE (Guth et al., 2016, 2018), are not activated in this study and thus are not discussed in the present paper. Heterogenous chemistry is treated in the stratosphere but not in the troposphere.

MOCAGE has 47 sigma-hybrid vertical levels, from the surface up to 5 hPa. The vertical resolution ranges from 40 m near the surface, to 400 m in the free troposphere, and 700-800 m in the upper troposphere and lower stratosphere. The vertical resolution, though not ideal, can be considered medium and enable a representation of the main dynamical characteristics of the UTLS (Miyazaki et al., 2010; Geller et al., 2016).

Since MOCAGE is an off-line CTM, it needs external meteorological forcing (temperature, pressure, humidity, wind, cloudiness and precipitation) from a separate meteorological or climate model. Long-range transport through advection is calculated with a semi-lagrangian scheme (Williamson and Rasch, 1989). Tracers transport through convection and diffusion are computed inside the model, according respectively to Bechtold et al. (2001) and Louis (1979).

Anthropogenic, biogenic, and biomass burning emissions are taken into account using precomputed inventories (choice of inventory is detailed in Sect. 2.2.2). Both anthropogenic and biogenic emissions are injected into the 5 lowest levels of the model. The injection fraction of the mass emitted $\delta r_e$ decays with levels $L$ above the surface ($L$ decreases with altitude): $\delta r_e(L) = 0.5 \delta r_e(L+1)$, ensuring a majority of emissions are still injected in the surface layer. Biomass burning injection scheme is detailed in 2.2.3. Nitrogen oxides ($NO_x$) emitted by lightning are computed according to Price et al. (1997).

The following processes concerning gaseous species are also represented in MOCAGE. Dry deposition is taken into account following Wesely (1989). Wet deposition is represented for both convective and stratiform precipitation, according to Mari et al. (2000) for convection, and the works of Liu et al. (2001) and Giorgi and Chameides (1986) for stratiform precipitation. Wet deposition is split into two processes, in-cloud scavenging (also known as rainout), and below-cloud scavenging (also known as washout).

MOCAGE has already been used to study various aspects of the chemical composition of the atmosphere, including the structure of the UTLS, and biomass burning plume transport. As part of the Chemistry and Climate Modelling Initiative (CCMI), it is used to study the impact of present and future climate on tropospheric and stratospheric chemistry, as well as transport of trace gases through the troposphere (Morgenstern et al., 2017; Orbe et al., 2017). Several studies on the extra-tropical UTLS and stratospheric intrusion have also been performed in Barré et al. (2013). It was also used in the framework

of the ChArMEx-GLAM airborne campaign, to study the origin of an observed CO anomaly in the upper troposphere sampled over the Mediterranean basin (Brocchi et al., 2017). Indeed, such CO enhancement was due to a biomass burning plume originating from both Siberian and North American forest fires. In order to correctly represent intercontinental transport of the plume in the previously mentioned study, the top of the fire emissions had to be set to an altitude of 10 km in MOCAGE to account for this extreme case of pyroconvection. Furthermore, Guth et al. (2018) highlighted the importance of the variability

of biomass burning emissions of pollutants and their direct impact to the basin aerosol budget over the Mediterranean. They showed that in 2013, fires in North America were the main contributor to the budget of primary organic aerosols over the Mediterranean basin. Our study explores in further detail the UTLS composition and its link to biomass burning emissions.

### 2.2.2 Simulation general set-up

MOCAGE dynamical forcing fields are taken from meteorological analysis performed by the ARPEGE model (Action de
Recherche Petite Echelle Grande Echelle) operated at Météo-France (Courtier et al., 1991). They are available at a 3-hourly time-step, and linearly interpolated at a 1 h time-step, to be coherent with the dynamical time-step of MOCAGE.

For anthropogenic emissions, MACCity inventory was used at a monthly resolution(Granier et al., 2011), complemented by the ACCMIP dataset for aircraft emissions (Lamarque et al., 2010). Concerning biogenic emissions, the MEGAN-MACC inventory is used (Sindelarova et al., 2014).

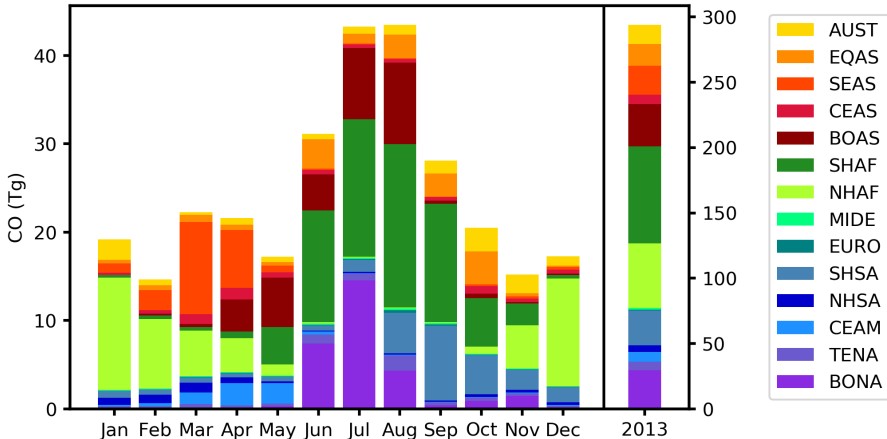

**Figure 4.** GFAS 2013 monthly and yearly emitted amounts of CO (in Tg) for the 14 geographical areas defined in Fig. 2.

MOCAGE is used in its global configuration, with a resolution of 1°longitude × 1°latitude (approximately 110 km × 110 km at the equator and 80 km × 110 km at mid-latitudes).

The simulation covers the whole 2013 year, with an additional spin-up period of three months (starting the simulation on the 1$^{\text{rst}}$ of October 2012). Hourly outputs from MOCAGE are used, as this is the minimum coherent time step of the model.

### 2.2.3 Biomass burning emissions set-up

In MOCAGE, biomass burning emissions are constrained using daily GFAS products (Kaiser et al., 2012). GFAS is an inventory based on the observation of the Fire Radiative Power (FRP) made by the MODIS instrument, on board of the Aqua and Terra satellites (Justice et al., 2002; Giglio et al., 2006). Based on the link made between FRP and fuel consumption, emitted amounts of trace gases and aerosols are calculated taking into account the land cover classification at the location of the observation. Emitted CO amounts are presented in Fig. 4 for the year of 2013, where emissions have been split according to the geographical areas defined in Fig. 2. With a total of 294 Tg of CO being emitted, 2013 biomass burning yearly emissions are below GFAS multiyear average. In the work of Petetin et al. (2018), where GFAS CO emissions were assessed over the same continental areas from 2002 to 2017, 2013 appears to be an average year for all regions, except for SHSA being below average and BONA emissions above average.

GFAS is averaged from its native resolution (0.1°longitude × 0.1°latitude) at MOCAGE resolution (1°longitude × 1°latitude) to set the location and amounts emitted for each day. But since no information on vertical repartitioning of the emissions of fires was available until v1.2 released in 2017, prescribed profiles of injection were used so far (see Fig. 5). In this approach the injection height was set depending on the latitude of the fire, even though it relies on other parameters like the type of fire. Injection fraction is then calculated above and below the injection height with an exponential decay. The top of the plume is set to 2 model levels above the maximum of injection fraction. The prescribed injection heights are 1 km, 2 km and 6 km,

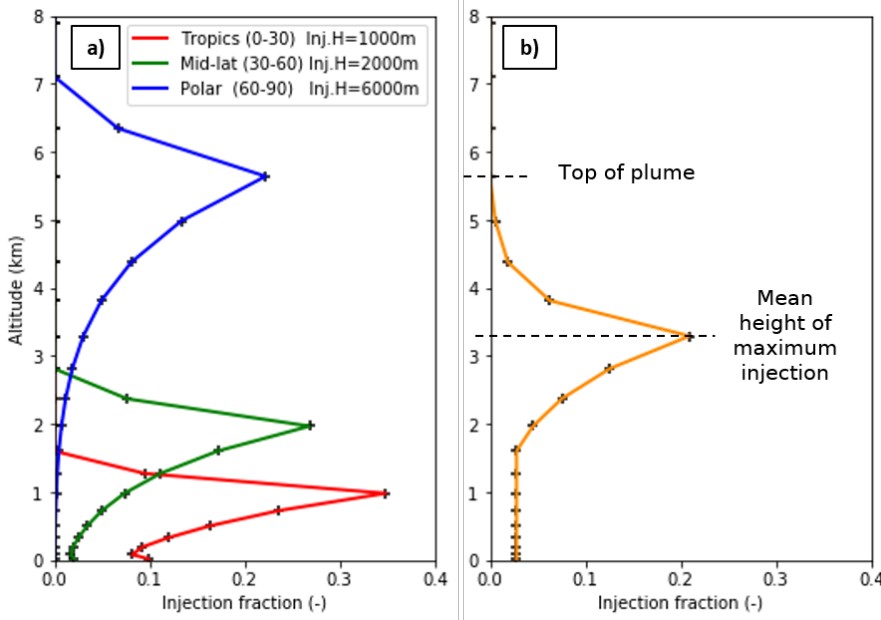

**Figure 5.** Injection profile schemes: a) BASE injection profiles with latitude dependent injection height b) INJH injection profile using GFAS plume rise parameters.

respectively applied for tropical latitudes, mid-latitudes and polar latitudes. These injection profiles were developed for the AMMA model inter comparison project (Williams et al., 2008, 2009) and are derived from the work of Lavoué et al. (2000). While it is a good approximation of the potential effect of pyroconvection on fire emissions, it does not take into account the difference between the active front of the fire and the smoldering areas, where pyroconvection rarely occurs. A first simulation

using this representation of fire emissions is performed, reference throughout this paper as the BASE simulation.

  As we want to investigate the role of the variability of injection height in the representation of biomass burning, a new way to set the injection profile in MOCAGE is developed. Instead of the prescribed injection height as a function of latitude, 2 products from GFAS v1.2 (Rémy et al., 2017) are used: the mean height of maximum injection (or injection height), and the plume top altitude. The injection height corresponds to the altitude at witch detrainment is at its maximum within the plume,

and thus injection of smoke at its maximum. The plume top altitude is simply the highest altitude reached by the plume. These products are derived from the measured Fire Radiative Power (FRP) from the MODIS instrument, and a Plume Rise Model (PRM) that takes into account the atmospheric state above the fire. They are available daily for most of GFAS fire pixels. These two plume rise parameters are used in MOCAGE to constrain the injection profile, as presented in Fig. 5. The injection profile is still set at its maximum at the GFAS injection height, and the GFAS plume top altitude is used instead of the fixed

two model levels above the injection height. A second simulation (referred to as INJH hereafter) using this new representation of biomass burning emissions is performed, identical in every other aspect to the BASE simulation. This new representation

**Table 2.** Simulations description summary

| Sim. name | BB emission inventory | Injection scheme |
|-----------|----------------------|------------------|
| NOBB | None | None |
| BASE | GFAS | Prescribed profile with zonal injection height |
| INJH | GFAS | Prescribed profile with injection height and top of plume from GFAS |

of fire injection will be compared to the one in the BASE simulation, using observations of biomass burning plumes from the IAGOS dataset, in section 3.

A third simulation (named NOBB) is performed where emissions from biomass burning are simply turned off (not just for CO, but all concerned trace gases). It will be used to study biomass burning impact on CO budget in the UT, presented in section 5. A summary of the three simulations and their specificities can be found in Table 2.

## 3    Improvement of biomass burning injection height in MOCAGE

### 3.1    Global evaluation of the plume representation

For the 220 plumes detected in the IAGOS database over 2013, CO mixing ratios measurements inside the plumes are compared with the MOCAGE model with the following methodology. Hourly results from the global model are interpolated on each plane trajectory at the measurement time and location to perform a collocated comparison. The interpolation is bi-cubic in space and linear over time. This results in a consistent dataset between the IAGOS CO measurements and each MOCAGE experiment, where each data value shares the same space and time coordinates.

To perform the IAGOS-MOCAGE comparison, the probability density function (PDF) for each dataset is computed , as well as box-and-whiskers plots. The results are first investigated for the plumes originating from the African continent and boreal forests, as they are the most sampled by IAGOS aircraft in 2013, and the main contributors to biomass burning emissions.

Looking at plumes originating from the African continent (either from NHAF or SHAF, see 2), both the PDF and the box plots are similar between IAGOS and MOCAGE for the two simulations. Both MOCAGE simulations well reproduce the CO mixing ratios distribution when compared to the IAGOS measurements, with a slight negative bias of -11 ppb between the means. The 95[th] quantile in particular is 26 ppb lower in both MOCAGE simulations compared to the IAGOS measurements. Comparing both simulations against IAGOS observations, it appears that using plume rise parameters does not improve the representation of the biomass burning plumes, as no significant difference can be seen between the two datasets. Indeed, this could be expected, because pyroconvection rarely drives fire emissions above 3-4 km into the troposphere for African fires. Looking at the vertical repartition of GFAS plume rise parameters in the INJH simulation in Fig. 7, it appears that the bulk of fire emissions occurs in the BPL, with a mean altitude of injection on around 2 km. This value is not far from the injection height for tropical latitudes in the BASE method of 1 km injection height. This difference is not significant because most of

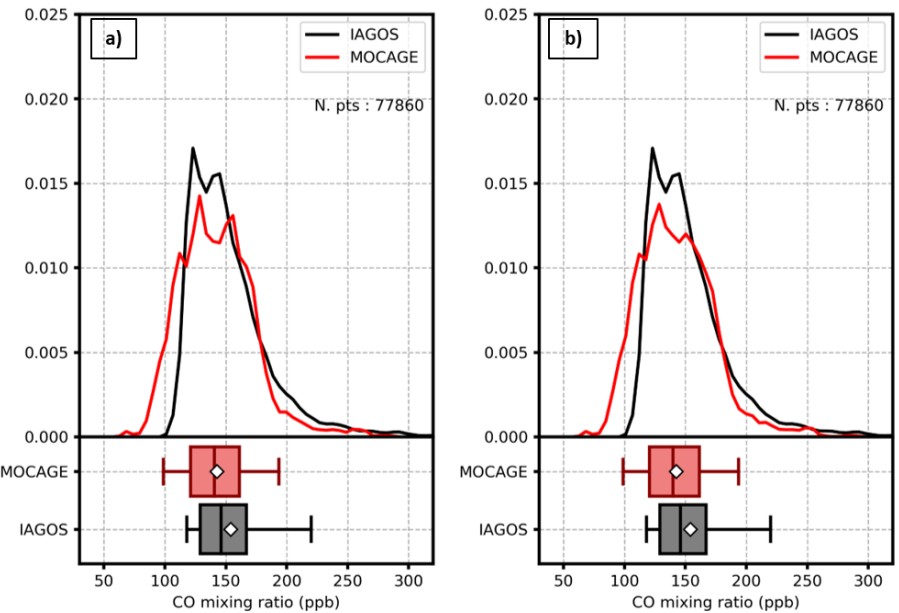

**Figure 6.** Probability density functions and box plots comparing IAGOS CO mixing ratios (in black) inside identified African Biomass burning plumes and corresponding MOCAGE simulated CO mixing ratios (in red) from the **a)** BASE simulation **b)** INJH simulation. Whiskers on box plots represent the 5th and 95th quantiles and the white diamond is the arithmetical mean.

emissions still occurs in the PBL and plumes that reach the cruise altitude of the aircrafts (10-11 km) in the tropical tropopause layer (TTL) are mostly transported thought deep convection. Hence, the injection profile difference on CO emissions altitudes is not noticeable at this range of altitudes.

However, looking at Fig. 8, where the same statistical comparison is performed for boreal biomass burning plumes, the PDF of CO mixing ratios in the BASE simulations shows a strong underestimation when compared to the IAGOS one. In particular, the strongest enhancements are not represented at all in MOCAGE, with a difference of more than 60 ppb for the 95th quantile between the BASE simulation and IAGOS CO mixing ratios. Using injection heights from GFAS improves the simulation results. The INJH repartition of simulated CO mixing ratios is a lot more similar to the observed one when compared to the BASE simulation. There is a slight negative bias of around 20 ppb between the modelled plumes and the measurements, but the strongest anomalies above 200 ppb are now represented. Still, a significant part of the simulated CO mixing ratios fall in the 50-100 ppb range, the background level of CO concentrations in the UT. This can be partly explained as a background negative bias in MOCAGE of about 10 ppb. It is also at least partially due to the numerical diffusion, in addition to the vertical resolution of MOCAGE, which is 700-800 m at the considered range of altitude. This improvement of MOCAGE results to represent upper tropospheric plumes might appear counter-intuitive regarding the fact that injection height is now on average lower in the INJH than in the BASE simulation. It comes from the fact that the variability in injection height is now correctly

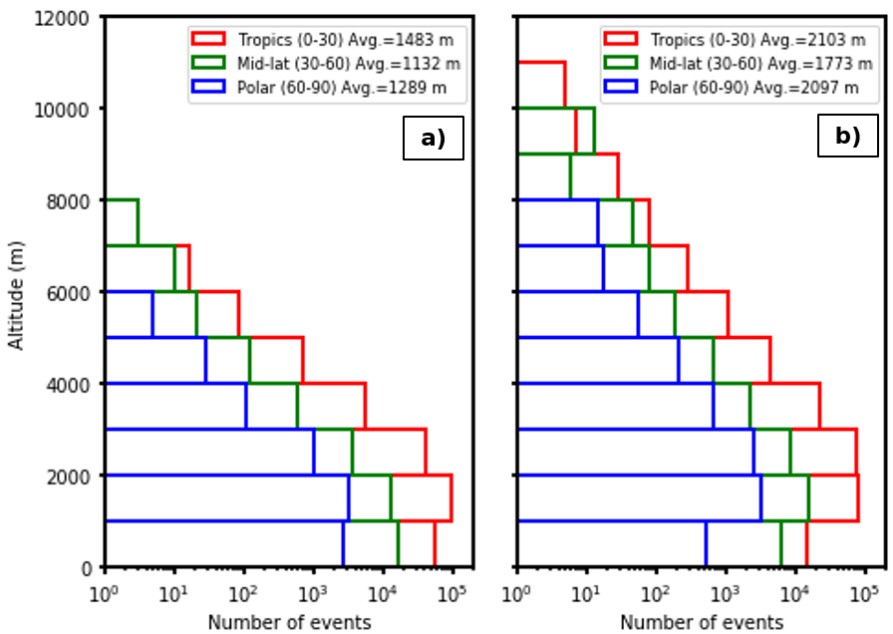

**Figure 7.** Vertical distribution of GFAS plume rise parameters over 2013 at a 1°longitude × 1°latitude resolution.**a)** Injection Height and **b)** Plume top altitudes

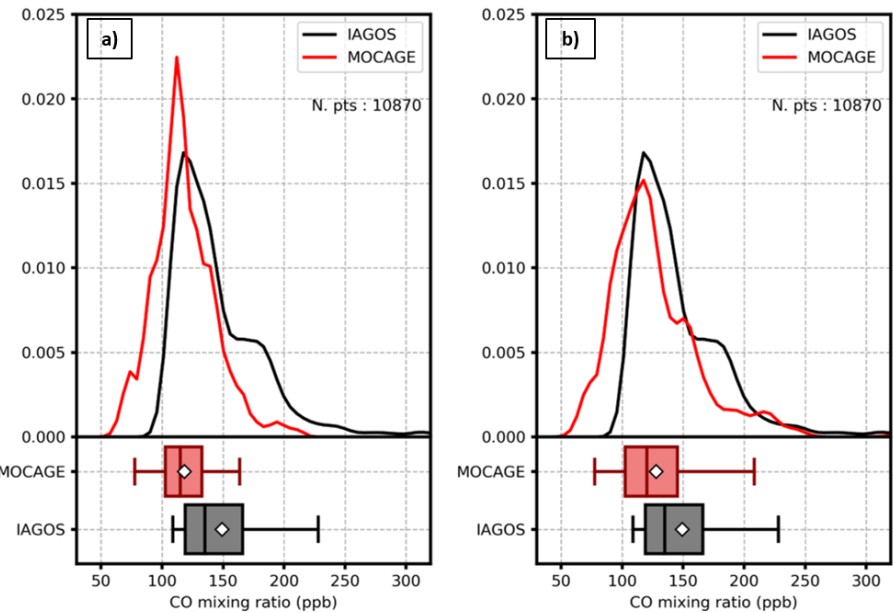

**Figure 8.** Same as Fig. 6, but for biomass burning plumes originated from boreal regions.

**Table 3.** Comparison of scores between MOCAGE simulation and IAGOS measurements for plume identified thanks to SOFT-IO

| Origin (N. plumes) | | BASE | INJH |
|---|---|---|---|
| | Sp. Corr. | 0.45 | 0.46 |
| AFR (95) | MNMB | -0.07 | -0.08 |
| | FGE | 0.17 | 0.17 |
| | Sp. Corr. | 0.60 | 0.67 |
| BOREAL (51) | MNMB | -0.21 | -0.15 |
| | FGE | 0.24 | 0.22 |
| | Sp. Corr. | 0.61 | 0.61 |
| OTHERS (74) | MNMB | -0.14 | -0.14 |
| | FGE | 0.23 | 0.23 |
| | Sp. Corr. | 0.54 | 0.55 |
| ALL (220) | MNMB | -0.10 | -0.10 |
| | FGE | 0.19 | 0.19 |

represented. We believe that this leads to a better representation of the actual pathways, either though direct pyroconvection or advection in the lower layer of the atmosphere and convection, that biomass burning emissions can take towards the UTLS.

The statistical comparison between the BASE and INJH simulation has also been performed for plumes originating from other regions than AFR or BOREAL (category OTHERS in Table 3), but since it shows no significant difference between the two, it is not presented here. Lastly, considering the dataset regardless of the origin of the plume (also not shown), African emissions contributes to more than half of the data points, hence conclusions are the same as for Fig. 6.

In order to complement the analysis, statistical scores have also been computed for each of dataset of plumes, for the two different MOCAGE simulations against the IAGOS biomass burning plume measurements. The chosen metrics are the Spearman's correlation coefficient, the Modified Normalized Mean Bias (MNMB) and the Fractional Gross Error (FGE). Those three metrics have been chosen to better represent an atmospheric chemistry model performance against in-situ data and have been used in recent studies in a similar way (Elguindi et al., 2010; Wagner et al., 2015). Each metrics calculation is detailed in the Appendix A. Spearman's correlation coefficient ranges from -1 (perfect anti-correlation) to 1 (perfect correlation). MNMB ranges from -2 to 2, with 0 being the target for no bias. FGE ranges from 0 to 2 with 0 being the target as well. The statistical evaluation of the simulations is presented in Table 3.

Looking at MNMB and FGE across the different origins, both scores are within acceptable range for each MOCAGE simulation. The MNMB is negative for all set of plumes, meaning that the model slightly under-represents CO enhancement inside biomass burning plumes. This can easily be explained as the model being diffusive, considering the size of each grid cell, as well as a general small negative bias of MOCAGE in the UT. Spearman's correlation is acceptable across the different

plumes origin, though not great either. This can be explained by the plumes being slightly offset spatially and/or temporally in the model, due to uncertainties in the dynamics as well as both the vertical and horizontal resolution.

For all origins except the boreal regions, metrics are the same regardless of the method of injection of biomass burning emissions. This is consistent with the comparison presented earlier in this section. For plumes originating from boreal regions, both the correlation and the MNMB are significantly improved using the injection height and plume top estimation from GFAS. Again, this is consistent with comments made on Fig. 8. It also brings the MNMB closer to values considered for other plumes, making the representation of biomass burning plumes in the model more consistent overall. It has also been verified that the INJH method of injection does not degrade MOCAGE performances outside of biomass burning plumes. Statistics for both simulation on the IAGOS database excluding measurements in biomass burning plumes are identical and thus not presented here. This is why only the INJH simulation will be validated globally in Sect. 4.

Although the original representation of biomass burning injection height in MOCAGE was giving fairly good results, using GFAS improves largely the ability of MOCAGE to forecast biomass burning plumes in boreal regions. This is because GFAS provides an estimate of the actual height of pyroconvection for each fire in these regions, and therefore captures the variability of injection heights. Combined with vertical transport processes, mainly convection, it allows the model to capture well boreal plumes.

## 3.2 Global impact of the plume representation

In order to investigate the impact of the new injection scheme (using GFAS plume rise parameters) on carbon monoxide distribution in the upper troposphere, we perform a global comparison between the BASE and INJH MOCAGE simulation. We treat monthly averaged results, but only August 2013 is shown hereafter as it is the month when the greatest difference between the two simulations can be seen.

In Fig. 9 is displayed the August 2013 CO vmr mean for both BASE and INJH MOCAGE simulations at 350 hPa, as well as absolute and relative differences between the two fields. At this pressure level (an altitude around 8 km), most of the difference between the two experiments is situated in the northern hemisphere, at mid to high latitudes, and especially above the Asian boreal forests. The 60°N latitude stands out as the demarcation line between different injection height (from mid-latitudes to high-latitudes) in the BASE simulation. It results in more CO being found at this altitude South of the 60°N parallel, up to 15 pbb, and less North, up to -25 ppb. It results in less CO being found for simulation INJH at this altitude than in the BASE simulation (up to -25 ppb) North of the 60°N parallel, and more CO (up to +15 ppb) South. This sharp discontinuity as been smoothed in the INJH simulation, as it had no physical source but was the result of an arbitrary demarcation in injection height.

It is also noticeable that no other area of the globe shows differences regarding the injection method used for biomass burning emissions. Despite finding injection heights and plume top altitudes reaching the upper troposphere all over the globe (see Fig 7), only the events in the Asian boreal forests seem to have emissions reaching the top of the troposphere with sufficient amounts to influence the monthly concentration of CO.

Knowing that most of the impact of the INJH method of injection is over the Asian boreal forests during the fire season (i.e. boreal summer), a zonal mean is created over the box defined in Fig. 9. This zonal mean is presented in Fig. 10, for the

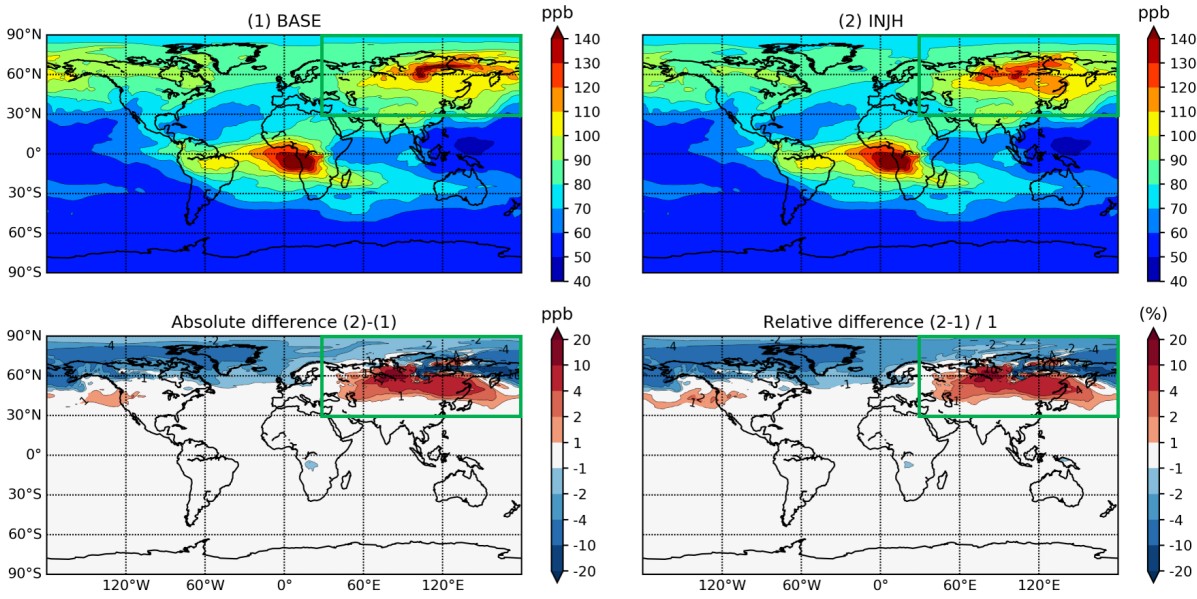

**Figure 9.** Monthly mean differences of CO mixing ratios between the BASE and IJNH MOCAGE simulations on 350 hPa iso-surface for the month of August 2013. The green box indicate the area used for the zonal mean.

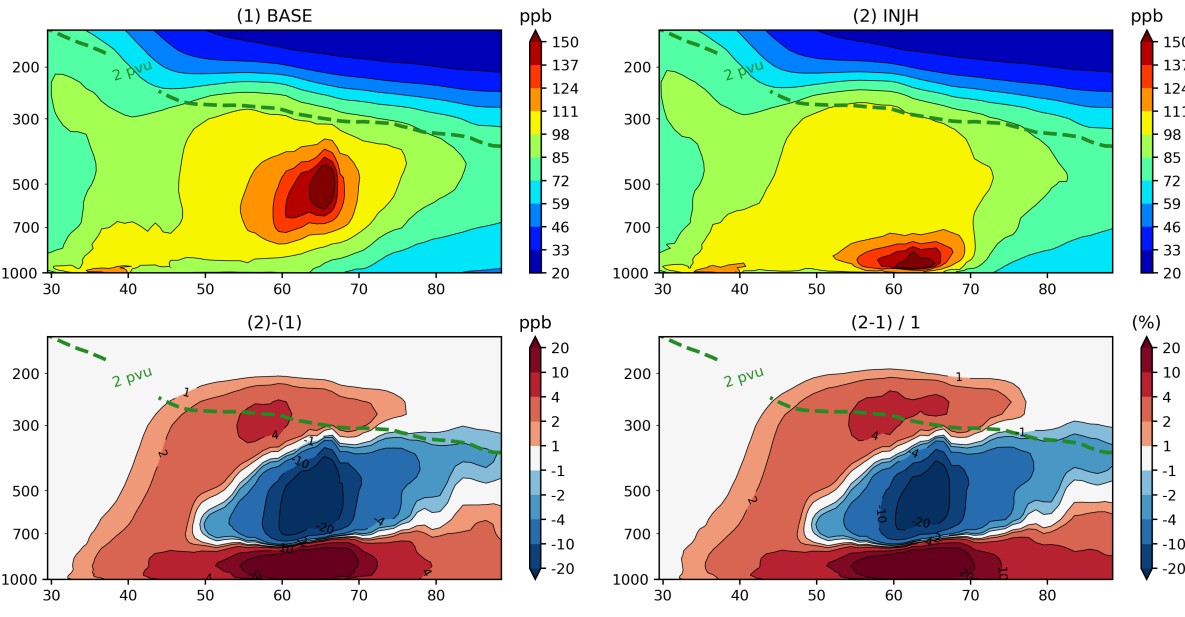

**Figure 10.** Zonal mean of CO mixing ratios for the month of August 2013. The mean is calculated between 30N-90N and 30E-180E. Doted green line indicates the 2 pvu iso-surface

month of August 2013 as well. In the BASE simulation, the maximum of CO mixing ratios is well defined and located in the free troposphere, around 600 hPa and between 60°N and 70°N. This is certainly the result of the fixed injection profile used in this method. While it allows CO to be injected in the free troposphere, it probably misrepresents its vertical distribution, as the mean GFAS injection height at these latitudes is around 2000 m (around 800 hPa). The difference between the INJH and

the BASE simulation further confirms this. In the INJH simulation, CO in much more abundant in the PBL than in the BASE simulation. The INJH also features an enhancement of CO mixing ratios above the dynamical tropopause identified as the 2 pvu iso-surface, probably is due to the few events of extreme pyroconvection that are characteristic of high-latitudes forest fires (Nedelec et al., 2005).

As shown in section 3.1, the INJH simulation showed better results for the simulation of biomass burning plumes originating

from boreal forests. This new method of injection of fire emissions also results in a more realistic repartitioning of CO over the Asian boreal region, but little to no impact elsewhere over the globe. The GFAS plume rise parameters have helped improving the representation of biomass burning emissions in MOCAGE, and this configuration of the model will now be used to assess the impact of biomass burning on the budget of CO in the upper troposphere.

## 4   MOCAGE evaluation against IAGOS measurements

Before assessing the impact of biomass burning on upper tropospheric CO, the INJH MOCAGE simulation is compared against the IAGOS database for the year of 2013. The objective is to validate the ability of MOCAGE to represent realistic CO in UTLS altitude range. For this purpose we calculate statistical scores for the IAGOS measurements made during the cruise part of the flights to avoid sampling the free troposphere and the PBL happening during ascent or descent of the plane. Statistical scores used are the MNMB and FGE described in Appendix A for the same reasons given in Sect. 3.1. The results are presented in

Table 4, where scores have been detailed for the three geographical regions defined in Fig. 1. Metrics are also being presented separately for the UT and the LS, the UT being considered between +15 and +75 hPa below the dynamical tropopause, and the LS between -15 and -150 hPa above it. In the CeAfr area, where the definition of the dynamical tropopause diverges, the 380 K iso-surface of potential temperature is used when the 2 PVU iso-surface becomes higher. In addition, the lower stratosphere is not sampled by the IAGOS planes in the tropics and thus scores are not presented for this area.

Overall the statistics indicate a good representation in MOCAGE of the CO mixing ratios both in the UT and the LS, with MNMB and FGE being close to zero in each of the regions. In both mid-latitude regions (EurNAt and cAsia), MNMB is slightly negative in the UT, and slightly positive in the LS. Correlation is good across all regions, except in MAM and JJA in EurNAt and cAsia. Except for this, scores do not appear to feature an annual cycle, with MNMB and FGE varying very slightly through the different seasons. MOCAGE seems to correctly represent both the upper tropospheric and lower stratospheric CO

mixing ratios at northern mid-latitudes, as well as the upper tropospheric CO over central Africa.

To complement the validation of MOCAGE behaviour in the UTLS, vertical profiles are computed using measurements during the ascent and descent phase of the aircraft. Airport clusters are formed by selecting nearby airports in order to take into account a larger number of flights. The airports considered in each cluster are described in Table 5, as well as the total

**Table 4.** Statistical scores comparing the INJH MOCAGE simulation against the IAGOS measurements in the three geographical regions defined in Fig 1. The annual mean of each score is written in bold, followed by the seasonal mean in brackets as follows (DJF | MAM | JJA | SON).

| Region | | MNMB | FGE | Correlation |
|---|---|---|---|---|
| NAtEur | LS | **0.07** ( 0.03 | 0.17 | 0.03 | 0.02) | **0.20** ( 0.16 | 0.24 | 0.22 | 0.14) | **0.67** ( 0.70 | 0.76 | 0.67 | 0.70) |
| | UT | **-0.12** (-0.04 |-0.16 |-0.13 |-0.11) | **0.15** ( 0.12 | 0.17 | 0.16 | 0.12) | **0.66** ( 0.62 | 0.39 | 0.49 | 0.56) |
| cAsia | LS | **0.14** ( 0.07 | 0.18 | 0.19 | 0.08) | **0.21** ( 0.16 | 0.24 | 0.25 | 0.15) | **0.70** ( 0.78 | 0.78 | 0.65 | 0.78) |
| | UT | **-0.10** (-0.04 |-0.20 |-0.09 |-0.11) | **0.16** ( 0.09 | 0.20 | 0.18 | 0.13) | **0.52** ( 0.71 | 0.55 | 0.40 | 0.67) |
| cAfr | UT | **-0.04** ( 0.06 | 0.20 |-0.06 |-0.11) | **0.15** ( 0.16 | 0.20 | 0.15 | 0.14) | **0.71** ( 0.70 | 0.89 | 0.58 | 0.89) |

**Table 5.** Description of used airport clusters for vertical profiles, as well as the total number of vertical profiles.

| Cluster's name | Airports | N. profiles |
|---|---|---|
| NAwest | Seattle, Vancouver | 215 |
| USlake | Chicago, Detroit, Toronto | 89 |
| USeast | New York City, Boston, Washington D.C., Philadelphia | 265 |
| France | Paris (CDG and ORY) | 318 |
| Germany | Frankfort, Düsseldorf, Munich | 1567 |
| MidEast | Dubai, Abou Dabi, Muscat, Riyadh, Kuwait | 158 |
| Windhoek | Windhoek (Namibia) | 215 |
| ChinaSE | Taipei, Hong Kong, Xiamen | 259 |
| Japan | Narita, Nagoya, Osaka | 196 |
| AsiaSE | Bangkok, Ho Chi Minh | 199 |
| IndiaS | Chennai, Bombay, Hyderabad | 210 |
| AfrW | Niamey, Ouagadougou, Bamako | 65 |
| NSAm | Cayenne, Caracas | 151 |

number of flights taking of or landing at each of these airports or cluster of airports. These profile are calculated by averaging measurements on fixed pressure levels (a total of 16 levels logarithmically spaced between the ground and the highest pressure reached during the ascent). Only measurements within 400 km of each airport of the considered cluster are used for the profiles, to keep the profiles representative of a geographical region.

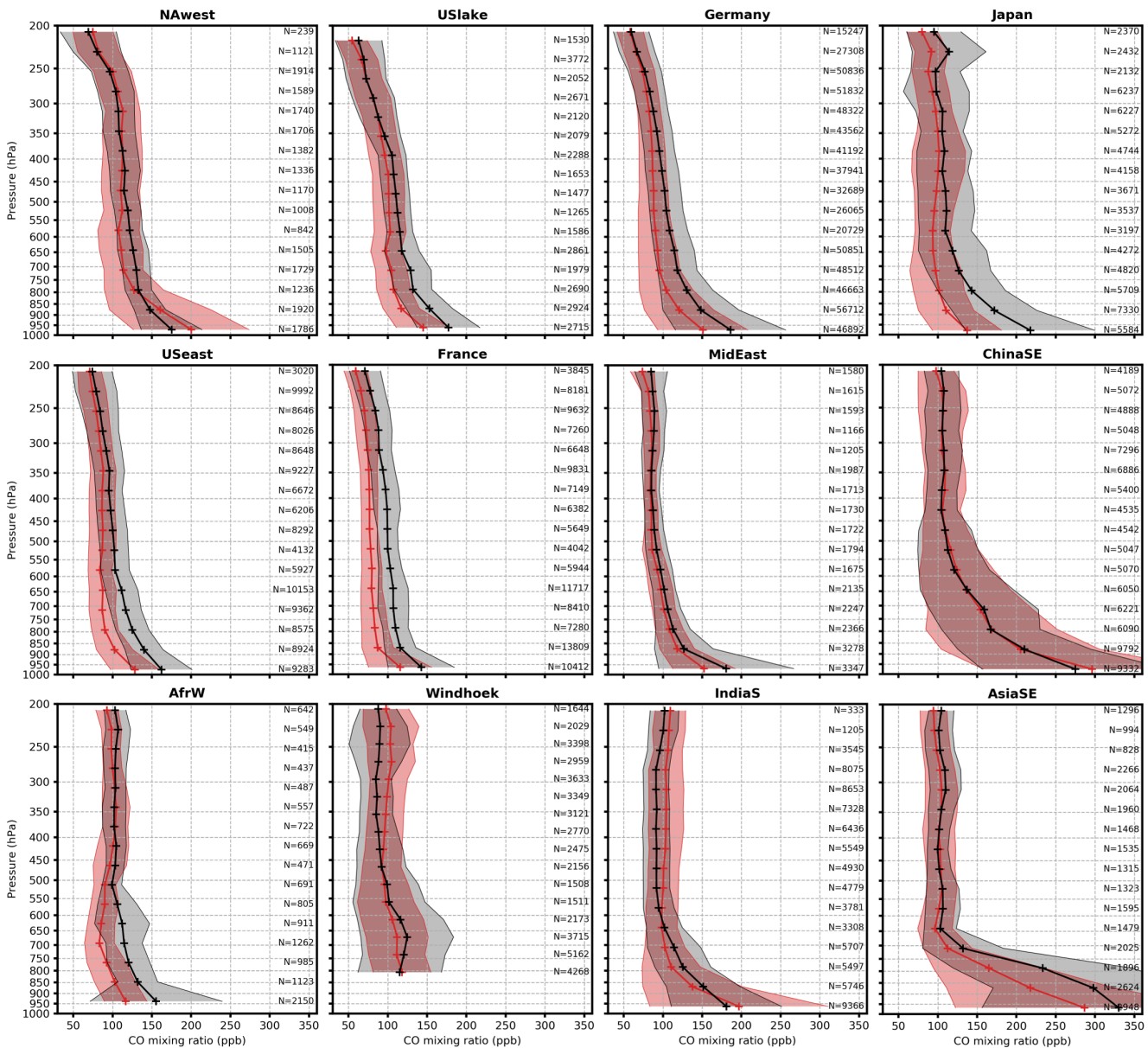

**Figure 11.** Comparison between IAGOS (in black) and MOCAGE INJH (in red) 2013 mean vertical CO profiles at the 12 airport clusters defined in table 5. The shaded grey and red areas represent respectively the standard deviation around the IAGOS and the MOCAGE mean. The number of data points used for each each pressure level mean is indicated on the right side of each profile.

The 12 annual mean vertical profiles are presented in Fig. 11. Profiles are presented from the ground to 200 hPa ( 12 km), but MOCAGE performances will mostly be discussed in the upper troposphere (300-200 hPa), which is the region of interest in this study. The same evaluation have been carried out seasonally, and are presented in the supplement to this article.

Looking at profiles representative of mid-latitudes (i.e. NAwest, USlake, USeast, France, Germany, and Japan), they show similarities when comparing MOCAGE CO mean mixing rations to IAGOS. Around 500 hPa, MOCAGE CO mixing ratios are underestimated by 10 ppb or more, but become closer to the IAGOS mean profile upper in the atmosphere, except for the France cluster. In this specific cluster MOCAGE keeps underestimating upper tropospheric CO by 10 ppb. It could be due to an overestimation of the oceanic influence of less polluted air in MOCAGE. At mid latitudes, both the vertical gradient and concentrations of CO in the free and upper troposphere are captured by the model.

For the six other clusters, all located within the tropics, only the troposphere is sampled as the tropopause is above any commercial flight levels in these regions, resulting in almost no vertical gradient in CO concentrations. In the AfrW, AsiaSE, MidEast and ChinaSE clusters, MOCAGE mean vertical profiles matches very well the IAGOS ones with a similar variability. At the Windhoek and IndiaS clusters however, MOCAGE tends to overestimate CO mixing ratios above 400 hPa by almost 20 ppb. MOCAGE mean profile still falls within the standard deviation around IAGOS mean profile. These overestimations of upper tropospheric CO could be a conjunction of an overestimation of surface emissions (especially for IndiaS, where MOCAGE overestimates surface CO), and an overestimation of convective transport, since both of these profile show a deficit of simulated CO in the PBL.

Finally, to evaluate in detail MOCAGE representation of CO vertical gradient in the UT and the LS, cruise altitude measurements are used to produce vertical profiles with a pressure coordinate relative to the dynamical tropopause. The annual profiles are presented in Fig. 12, for the EurNAt and cAsia regions. The transition from the UT to the LS can clearly be seen on the IAGOS mean profile, with CO mean mixing ratios falling from 90-100 ppb to below 50 ppb above the dynamical tropopause. This transition is also observable in the MOCAGE mean profile, but with a weaker vertical gradient. This results in a slight underestimation in the UT and a slight overestimation in the LS of CO mixing ratios, which was already seen in the scores presented in Table 4. The variability observed, represented in the figure by the deviation around the mean, is correctly captured by MOCAGE as well. Similarly to the vertical profiles of Fig. 11, this evaluation has been preformed for the four seasons of the year, and is presented in the supplement to this article.

To summarize, MOCAGE chemistry transport model, when compared to IAGOS in-situ observations, is able to reproduce realistic CO mixing ratios from the free troposphere up to the upper troposphere and the lower stratosphere.

## 5 Global impact of biomass burning on upper tropospheric carbon monoxide

### 5.1 Global contribution

In order to assess the impact of biomass burning emissions on global CO concentration in the upper troposphere over the globe, the INJH and NOBB MOCAGE experiments are compared. As we want to investigate the contribution of biomass burning emissions on upper tropospheric composition, an upper tropospheric layer is defined in the MOCAGE simulation results. This layer is defined relative to the dynamical tropopause, ranging between -15 hPa to -65 hPa below the 2 PVU iso-surface, as done in previous studies analysing the UTLS with IAGOS data (Thouret et al., 2006; Cohen et al., 2018). Near the equator, where the potential vorticity rises exponentially, the dynamical definition is replaced by the 380 K potential temperature iso-

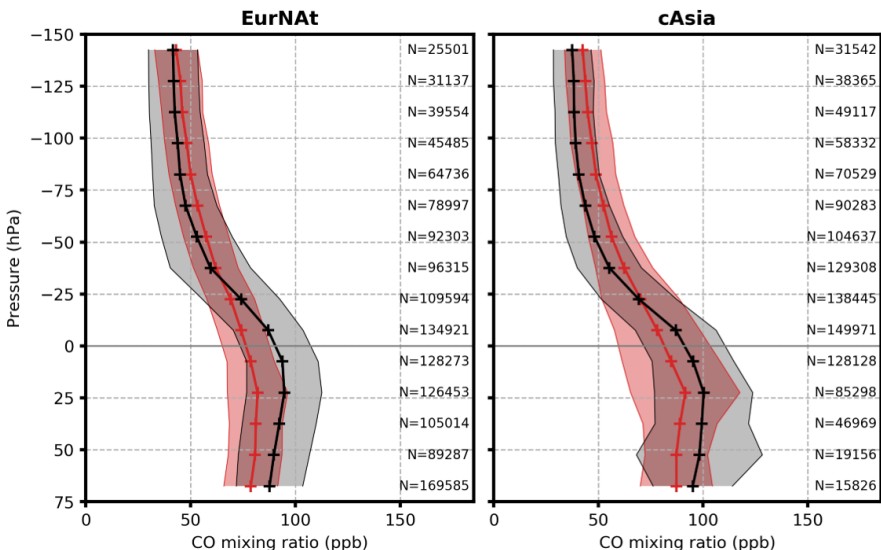

**Figure 12.** Comparison between IAGOS (in black) and MOCAGE INJH (in red) for the 2013 mean vertical CO profiles in the two geographical areas EurNAt and cAsia defined in Fig. 1. Pressure is relative to the dynamical tropopause, defined as the 2 PVU iso-surface. The shaded grey and red areas represent respectively the standard deviation around the IAGOS and the MOCAGE mean. The number of data points used for each each pressure level is indicated on the right side of each profile.

surface whenever the 2 pvu iso-surface is higher. The MOCAGE results are then averaged in this layer. The estimation of the contribution of biomass burning is calculated by subtracting the results of the NOBB simulation from the INJH one.

In Fig. 13 are displayed the MOCAGE annual mean CO mixing ratios for 2013 in the upper troposphere for the INJH and the NOBB experiments, as well as the calculated difference between the two. A similar map as the INJH but with IAGOS data could not be produced as a year of IAGOS flight is not enough to establish a representative climatology. However, the map looks consistent with the 15 year IAGOS CO climatology shown in Cohen et al. (2018). It appears that biomass burning emissions contribute globally to upper tropospheric CO. The averaged contribution can be estimated around 8 ppb for 2013, which is between 10 and 15 % of the CO mixing ratios at this altitude. Looking at the difference between the two simulations, 2 regions stand out with above average contribution of biomass burning, namely equatorial Africa and boreal latitudes, with locally more than 15 ppb of CO of difference between the two simulations. This is expected as these two regions represent respectively 42 % and 20 % of CO emissions from biomass burning in 2013, and transport pathways above these regions also favours injection of CO in the upper troposphere (Huang et al., 2014, 2012; Liu et al., 2013). As mentioned in Sect. 3.1, African emissions are transported to the upper troposphere thanks to deep convection, while a fraction of Boreal fires emissions are transported directly high in the troposphere by pyroconvection.

In Fig. 14 is presented the monthly mean evolution of the absolute difference of UT CO mixing ratios between the INJH and the NOBB simulations. It is worth noting that at all times of the year, biomass burning contributes to simulated UT CO mixing

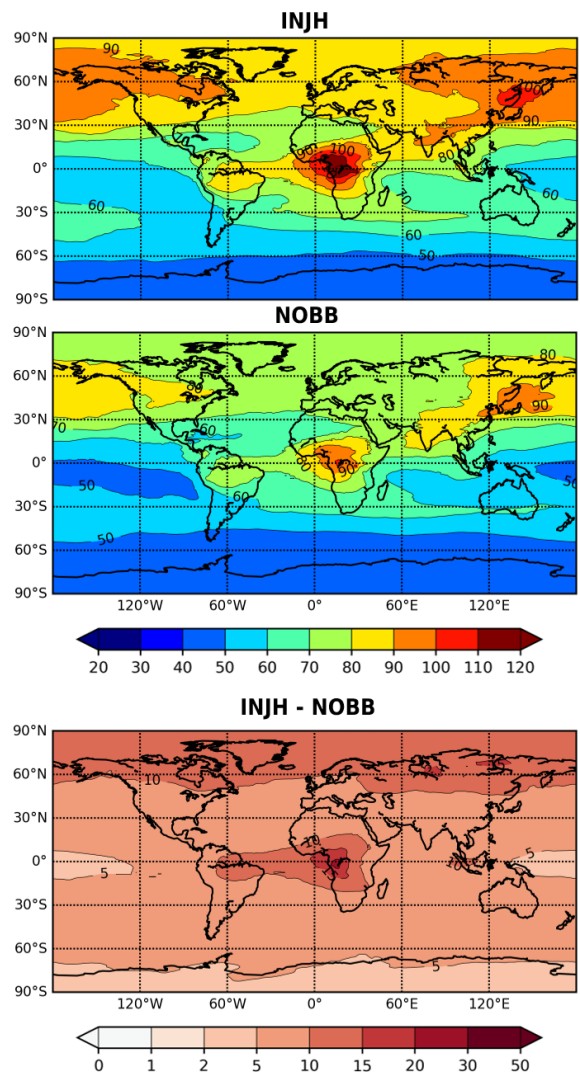

**Figure 13.** Map of 2013 upper tropospheric mean mixing ratios of CO for the INJH and NOBB simulations, as well as the absolute difference between the two.

ratios, by more than 5 ppb. In an expected way, the contribution of biomass burning is higher directly above the most active open fire areas. Equatorial Africa is impacted all year round, north of the equator from December to May, and south of the equator from June to November, with a contribution of more than 15 ppb of CO. Boreal fires have the strongest impact during the two most active months of forest fires (July and August), with an enhancement of up to 67 ppb of CO at boreal latitudes,

5    which is approximately half of the CO in this region. Other regions being impacted are the Amazon forest during its fire season

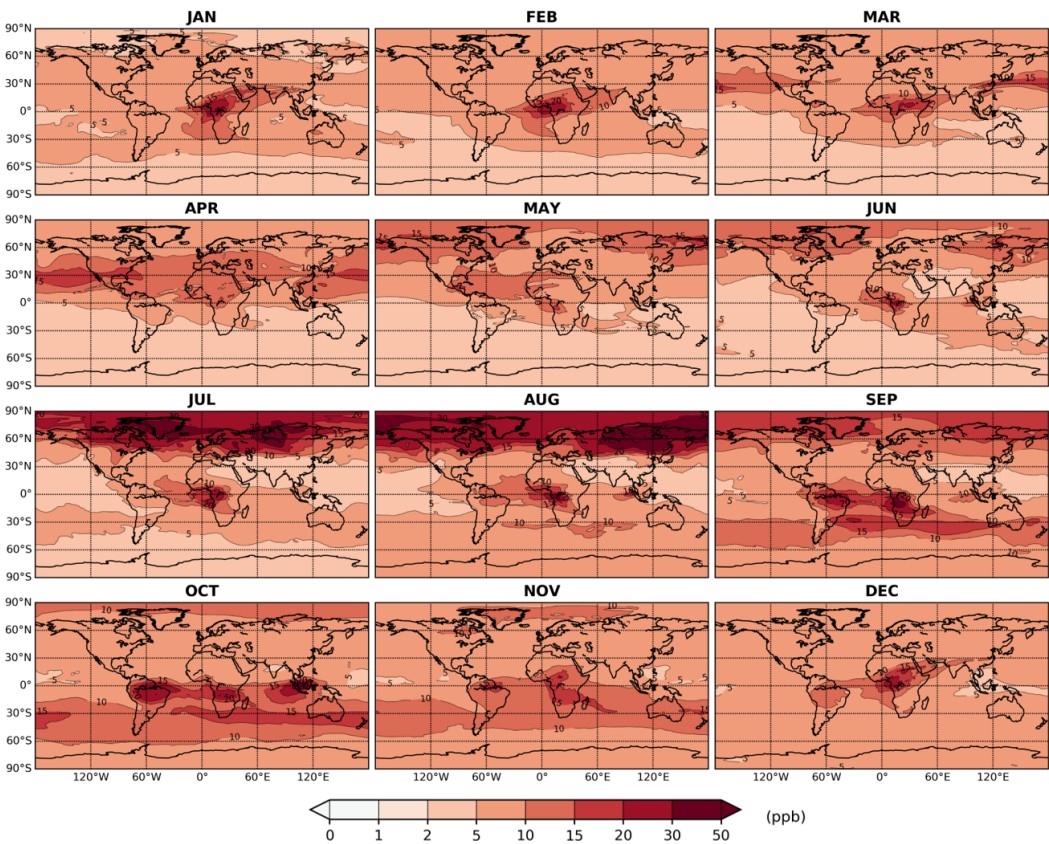

**Figure 14.** Maps of the 2013 monthly mean difference in upper tropospheric CO mixing ratios in ppb between the INJH and the NOBB MOCAGE simulations.

(September to November), and Indonesia in October. Even though the strongest impacts are local, the relatively long lifetime of CO (approximately 2 months) makes biomass burning emissions impacting UT air composition on a global scale.

## 5.2 Sensitivity to regional emissions

In order to clarify how each region affects the upper tropospheric CO, different variations the INJH simulation were performed. In each one, the biomass burning emissions were activated only for one region of the globe with significant amount of CO emitted from biomass burning (see Fig. 4). The chosen regions are the following :

- AFR (NHSA and SHSA), with 124 TgCO in 2013 (42 %)
- BOREAL (BOAS and BONA), with 62 TgCO in 2013 (21 %)
- AMAZON (NHSA and SHSA), with 32 TgCO in 2013 (11 %)
- EQAS, with 17 TgCO in 2013 (6 %)

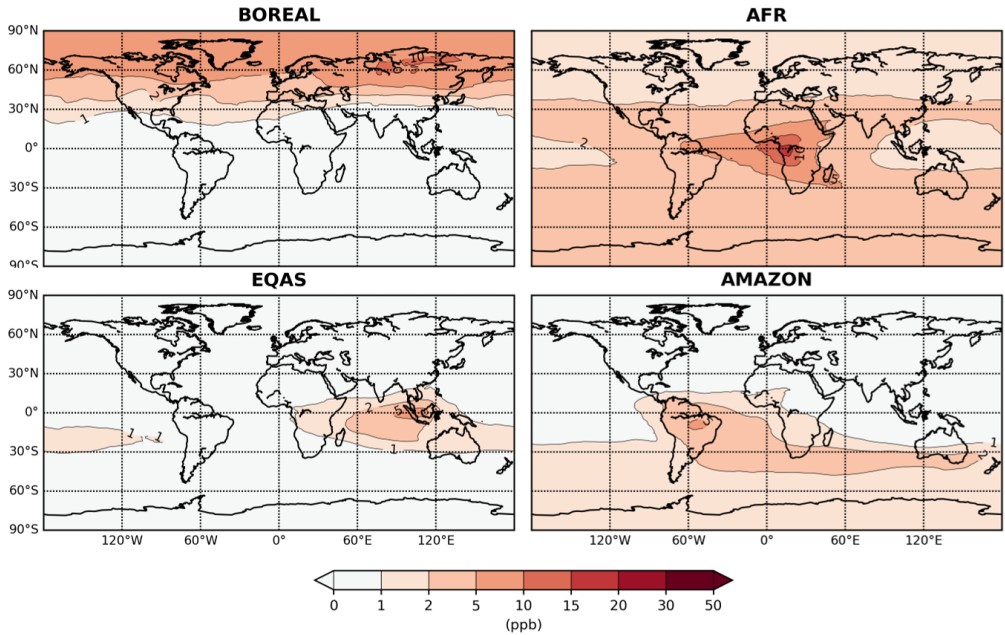

**Figure 15.** Maps of upper tropospheric CO yearly mean differences between the four sensitivity tests of the INJH simulation and the NOBB simulation, in ppb.

The resulting yearly means of upper tropospheric CO are presented in Fig. 15. As expected, impacts are greatest for African and Boreal fire emissions, since both of these regions are the main contributors to annual emissions overall. Due to atmospheric mixing, African emissions have a wider and global impact, with an enhancement between 2 and 5 ppb on upper tropospheric CO. Boreal fire emissions also have a strong impact with an enhancement of 5 to 10 ppb, but limited to boreal latitudes. Emissions from the Amazon forest, despite being slightly lower than usual in 2013 (Petetin et al., 2018), impact the upper troposphere by more than 5 ppb directly above the regions, but also a good part of the southern hemisphere through general circulation. Emissions over the maritime continent (EQAS) mostly have a local impact on the upper troposphere, for around 5 ppb of CO.

Three additional simulations were performed where emissions were limited to the SEAS or AUST regions, as well as a last one for the 5 remaining regoins (TENA, CEAM, EURO, MIDE and CEAS), since they account for similar amount of CO emissions in 2013 (respectively for 22 Tg, 15 Tg and 23 Tg) as the EQAS region. The results are not shown since no noticeable effect was seen on upper tropospheric CO. However they allowed for a verification of this method by adding each CO contribution field and comparing them to the one presented in Fig. 13. Yearly mean results (not shown) matched closely within 1 ppb. Australian fires, despite being active almost all year long, might not have fast enough transport pathway to the upper troposphere, as convection is less prevalent than in equatorial Asia (Liu et al., 2013), and fires are more spread out geographically. SEAS fire emissions on the other hand, despite being more important than in EQAS, do not impact the upper

troposphere. This is probably due to the time of occurrence in 2013 of the fires. SEAS fires mostly occur from February to May, before the beginning of the south asian monsoon, whereas EQAS fires occurs from June to October, covering the whole monsoon season, when fast vertical transport is the most effective through deep convection. In conclusion, biomass burning emissions contribute globally to the upper tropospheric CO budget. The impact is stronger above the areas with the strongest fire emissions, but is non negligible elsewhere around the globe due to atmospheric transport.

## 6    Summary and conclusions

Carbon monoxide is one of the key trace gases in the atmosphere, and one of its major sources is biomass burning. This study aimed at improving the understanding of the contribution of biomass burning emissions to upper tropospheric carbon monoxide, using global simulations performed with MOCAGE chemistry-transport model. We have chosen a single year simulation, as a compromise between the need to study seasonality of biomass burning emissions and computational costs as simulations were carried out globally at a fine resolution and with frequent outputs. The year of 2013 was chosen as it was the most recent year with the largest number of validated IAGOS flights.

Firstly, GFAS latest products (injection height and plume top altitudes) were used in MOCAGE to improve the injection scheme for biomass burning emissions (INJH simulation), as they are expected to better represent the characteristics of wildfire plumes and injection in the upper part of the troposphere due to pyroconvection for a selection of large fire events. To validate the results of this simulation, IAGOS in-situ observations were used, in addition with SOFT-IO ancillary data on CO anomalies source attribution. Thanks to the SOFT-IO contribution estimations, 220 plumes were selected from the IAGOS flights of 2013, so as to validate the use of GFAS injection height and plume top altitude in MOCAGE. Results were analysed accordingly to the region of origin of the plume. The original plume representation in MOCAGE performs fairly well. Nevertheless, the new injection scheme greatly improves the modelled plumes from boreal wildfires. It was also shown the importance of capturing the variability of pyroconvection on CO vertical distribution at northern high latitudes, but not elsewhere, despite higher occurrences of wildfires. To go a step further, it would be interesting to investigate the sensitivity to the shape of the injection profile of the vertical distribution of CO. Based on the results of this study and previous ones that have shown that the importance of the transport through pyroconvection has been underestimated (Fromm et al., 2019), it appears that the use of products such as GFAS injection height can offer improvement in the representation of biomass burning plumes in atmospheric models, especially for mid to high latitudes.

Secondly, the INJH simulation was evaluated using all the available CO IAGOS measurements of 2013 before being used to study the contribution of biomass burning to upper tropospheric CO. The extensive IAGOS database has been used in two ways :

– Vertical profiles at different airport clusters were used to validate CO levels as well as its gradient in the free and upper troposphere at specific locations.

– Overall statistical scores and vertical profiles calculation in the UT and LS confirmed MOCAGE is well suited to study upper tropospheric CO.

Finally, a global estimation of biomass burning emissions contribution to upper tropospheric CO is given, by comparing different simulations in which biomass burning emissions were activated or not in different regions of the globe. On average the contribution is 8 ppb in 2013, but gets up to 20 ppb above the regions with highest emissions (i.e. equatorial Africa and the boreal forests). Other regions where emissions have a significant impact are the Amazon forest and equatorial Asia. Overall, the amount of emitted CO seems to be the primary factor driving the upper tropospheric contribution, followed by the existence of fast vertical transport above or close to the burning areas (either deep convection or pyroconvection).

The conclusions of this work are of course limited to the year of 2013. As emissions from biomass burning this year are on the lower end, estimations made in this paper are expected to be as well. It would be interesting carry out similar analysis for years with high fire activity, or more generally investigate the role of inter-annual variability of fire emissions by extending the simulations. IAGOS database would still be valuable as it spans over more than 15 years for CO measurements. Before this, the first step will be to extend the analysis of the numerical simulations used in this paper, looking at wildfires impact on upper tropospheric chemistry. The ozone production especially will be investigated since CO is its main precursor at this altitude.

*Data availability.* The IAGOS time series (https://doi.org/10.25326/06), vertical profiles (https://doi.org/10.25326/07) and the SOFT-IO ancillary data (https://doi.org/10.25326/3) used in this study can all be found on the IAGOS data portal (https://doi.org/10.25326/20)

## Appendix A: Statistical score used for evaluation

The metrics used in this study for evaluation of modelled variables against in-situ observations are the modified normalized mean bias (MNMB), the fractional gross error (FGE), as well as Pearson's linear correlation coefficient and the Spearman's rank correlation coefficient. The MNMB and the FGE have been chosen over the standard mean bias (MB) and root mean square error metrics (RMSE),as they have been prescribed for atmospheric air composition studies where differences between a model and observations can be larger than for other meteorological fields (Seigneur et al., 2000).

For a dataset of $N$ observations points $o_i$ and co-located modelled value $f_i$, the MNMB is defined as :

$$MNMB = \frac{2}{N} \sum_{i}^{N} \left( \frac{f_i - o_i}{f_i + o_i} \right) \tag{A1}$$

It ranges from -2 to 2 and behaves symmetrically in regards of underestimation and overestimation.

$$FGE = \frac{2}{N} \sum_{i}^{N} \left| \frac{f_i - o_i}{f_i + o_i} \right| \tag{A2}$$

The FGE ranges from 0 to 2, and unlike the RMSE it does not overweight data points with the highest observed values within a given dataset.

In order to asses the model capability to reproduce the observed spatio-temporal pasterns, two types of correlation coefficient were used. Pearson's linear correlation coefficient is defined as follows :

$$r = \frac{1}{N} \frac{\sum_i^N (f_i - \overline{f})(o_i - \overline{o})}{\sigma_f \sigma_o} \tag{A3}$$

with $\overline{f}$ and $\overline{o}$ the mean value of the forecast and the observed values, as well as $\sigma_f$ and $\sigma_o$ the associated standard deviation.

This coefficient ranges between -1 for a perfect anti-correlation to 1, a perfect correlation. However, it assumes a linear relation between the forecast and the observations. This is not always the case for a global atmospheric composition model, at this is why Spearman's rank correlation coefficient $r_s$ has also been used. It is defined as the following :

$$r_s = \frac{cov(rg_f, rg_o)}{\sigma_{rg_f} \sigma_{rg_o}} \tag{A4}$$

where $rg_f$ and $rg_o$ are the rank of the forecast and observed values within the dataset, with $\sigma_{rg_f}$ and $\sigma_{rg_o}$ being their associated

standard deviations. Spearman's rank correlation is also bounded between -1 and 1, but does not make the assumption that the forecast and the observed values are linearly related. It particularly is valuable for the study of heavy-tailed distribution of observations, or distributions where strong outliers are expected.

*Author contributions.* MC, VM, VT and BJ designed the study and the model experiments. Simulations were carried out by MC with help from BJ. The IAGOS and SOFT-IO data were provided by VT and BS, with guidance on how to use them. Manuscript was written by MC

and reviewed by VM, and commented, edited and approved by all the authors.

*Competing interests.* The authors declare that they have no conflict of interest

*Acknowledgements.* The authors acknowledge the strong support of the European Commission, Airbus, and the airlines (Lufthansa, Air-France, Austrian, Air Namibia, Cathay Pacific, Iberia, and China Airlines so far) who carry the MOZAIC or IAGOS equipment and have performed the maintenance since 1994. In its last 10 years of operation, MOZAIC has been funded by INSU-CNRS (France), Météo-France,

Université Paul Sabatier (Toulouse, France), and Research Center Jülich (FZJ, Jülich, Germany). IAGOS has been additionally funded by the EU projects IAGOS-DS and IAGOS-ERI. We also wish to acknowledge our colleagues from the IAGOS team in FZJ, Jülich, for useful discussions. The MOZAIC–IAGOS database is supported by AERIS. The authors also thank the Midi-Pyrénées region and Météo-France for funding Martin Cussac's PhD.

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
