# Peer review of "The impact of biomass burning on upper tropospheric carbon monoxide : A study using MOCAGE global model and IAGOS airborne data."

_Atmospheric Chemistry and Physics, 2019_

## Referee Comment (RC1) · Anonymous Referee #1 · 17 Feb 2020

Review of **The impact of biomass burning on upper tropospheric carbon monoxide : A study using MOCAGE global model and IAGOS airborne data** by **M. Cussac et al., ACPD 2020**

This manuscript provides a nice study on the impact of biomass burning injection height parameterization on upper tropospheric CO, and the general contribution of biomass burning emissions on the upper tropospheric CO burdens, by using information on the injection height of biomass burning emissions as available from the GFAS product, the SOFT-IO tools for attributing emission sources, and the MOCAGE CTM.

The study is overall clear, and well-designed. Nevertheless, I have several concerns which the authors should address before this can be accepted for publication.

1. The structure of the manuscript is not ideal. I was expecting a general model validation of (upper-)tropospheric CO near the start, but this is only covered Sec. 4.1. I suggest to shift this to Sec. 3, e.g. just before Sec. 3.1
2. Although the evaluation metrics can well be defended (pdfs, MNMB, FGE, r), the data-aggregation is not fully suitable. There are three instances for this.
   a. when evaluating the impact of injection height in Table 3, I believe the authors should equally evaluate the improvement (if any) against IAGOS measurement that do not contain biomass burning contribution. I acknowledge that this are many more instances, (as already seen from Fig. 3) which implies that the changes are expected to be small. But in any case it'd be a good to check if there is no degradation, and hopefully an improvement, in this respect.
   b. Likewise, in Table 4 the authors present a general overview of the statistical scores in the UTLS, in terms of MNMB, FGE and r. But as the authors discuss, there are large gradients in CO in this region, which makes it hard to judge from these statistics to what extend these are captured. Unfortunately this is also not really visible from Fig. 13, which overall focusses on the free troposphere. Therefore I suggest that the authors also present metrics specific for below and above the tropopause, using the 2 PVU metric + / - a pressure range, e.g. as the authors already specify on Page 21, line 5.
   Additionally a figure presenting the vertical profile of CO in the UTLS region, ideally on the vertical axis the pressure level with respect to the tropopause altitude, would be very instructive.
   c. Finally, for Figure 13, it'd be good to discuss (or even present) similar evaluations, but specific for the four seasons, (or at least for summer and winter seasons), considering that there is a rather large seasonal cycle in tropospheric CO, and likewise variations in general CTM abilities to capture this, which would be good to quantify.
3. Although the manuscript is overall clearly written, there are still many issues. The introduction lacks many references, many acronyms are not defined, sentences are often not in proper English, and there are many typo's.
4. Finally, the authors emphasize the impact of pyroconvection throughout the manuscript, but an important impact of the revised injection height parameterization appears that overall much more is injected at lower altitudes compared to the reference configuration, whereas not even a single event where a likely pyroconvection has been sampled by IAGOS, is assessed in a little more detail. This makes the reference to this process a little out of balance.

**Other general comments**

P2

L25: To put the emission numbers in perspective, can the authors provide an estimate of the secondary CO production from VOC and CH4 oxidation?

L29: GFAS, GFED, IS4FIRES: Missing references.

P3: L9: The authors quickly argue that the use of airborne campaigns is limited. This is a very short conclusion. I think this deserves a little more elaboration, e.g. including a few studies based on campaign data, and/or reporting on some interesting findings.

P8.

L2'both anthropogenic and biogenic emissions are injected into the 5 lowest levels' . Why are biogenic emissions not injected at the surface layer?

L3: 'with an exponential decay': what do you mean?

L10-L23: in this section I am missing a reference where the overall performance of MOCAGE tropospheric chemistry is given; now bits and pieces are given from other references. This links a bit to my request to start Sec 3.1 with a general evaluation of tropospheric CO, if there is no suitable general evaluation reference existing.

P12, L1-L4: The authors show that in simulation INJH MOCAGE captures better the observed events with high CO (particularly those > 200 ppb), which they attribute to the higher injection height for the largest fires. However, from Figure 7 it is actually clear that injection of majority of fire emissions is now actually at much lower altitudes than before, which could possibly imply that for background conditions the MOCAGE now shows a larger negative bias in the UTLS than before? Could the authors comment?

In this line of thinking, to evaluate purely the effect of the injection height, it could be interesting to correct for the effect of a potential background model bias in CO, which also interferes with the evaluations shown in Figures 6, 8 and 9, which intend to focus on the effect of injection height, not the effect of potential model biases.

L7: "under-represents the CO enhancement": I think in this configuration this should read 'CO amount', as the authors did not actually compute the enhancement (see previous comment)

L8: "diffusive": I think this is only part of the explanation. Please consider general model biases. E.g. here it would be useful to directly refer to a corresponding evaluation for against IAGOS observations where likely fire emission contributions are excluded.

P15

L1-4 "to summarize…" I think these two sentences contradict each other. Either the original injection profile is good, or the GFAS product is good. Also the authors currently only focus on the impact in the UTLS. Although interesting, the largest impact of a different injection height might actually be for plumes at lower altitudes, where profiles on average look very different.

P18, Table 4: The overall small MNMB may not be very meaningful considering the large vertical gradients in this altitude range. Therefore I would appreciate if the authors could also compute the metrics for below and above the tropopause separately.

L15. 'MOCAGE seems to correctly represent the transition..' I find this difficult to judge. Here A figure showing the vertical profiles in CO with respect to the tropopause level would be very useful.

P19,

L18: 'realistic UTLS' -> 'realistic free tropospheric' ?

Figure 13: It'd be good to show this kind of evaluations for the different seasons separately. (or alternatively in the supplementary material)

P25

L15: "as well as its gradient in the UT". First of all, the vertical profiles as presented so far were not very clear in presenting the gradient in the UT. Secondly, it is not clear what is concluded from this evaluation.

**Specific comments**

Abstract: Please use present or past tense consistently.

L6: The use of these GFAS products lead**s** to improved MOCAGE skill to simulate fire plumes originating from boreal **forest** wildfires.
L10: 'as the previous one': which 'one'?
L11: database -> observations
L13: were -> where;  … biomass emissions were toggled …

Throughout the manuscript: Please check where you use 'quantity': I think it is better to use 'amount'

P2:
L5: great -> large
L8: focus is made here -> here we focus

L11: Methane (CH4) **and VOC** …
L13 'carbon dioxide'
L14: 'due to their shorter lifetime': this is not generally true. E.g. C2H6 has a lifetime of approx. two months

L23: 'in the **free** troposphere' ?
L23: since the occurrence of this phenomenon …

L34: class-> classification. Also good to add reference here?
L35: Their intensity: Unclear which intensity is meant here.

P3:

L1: 'pyroconvection can occur': please add references here.

L3: sensibility-> sensitivity?

L5: The authors refer to IASI, but the reference Deeter et al. is for MOPITT. Please resolve this inconsistency

L6: 'lack vertical resolution in UTLS': I think this is not only in UTLS, but generally?

L7 and L12: O3 -> $O_3$

L8: On **the** one hand…

L13: 'most likely source' -> 'most probable sources'?

L14: 'pathways' : you mean transport pathways, or chemical pathways.

L15/16: Suggest to change to: "This is why the analysis of large datasets require other kinds of additional information.."

L16: "Lagrangian backward transport calculation": add reference?

L20: …ability of global simulations performed by the MOCAGE … . Also please add reference to MOCAGE, and consider explaining ACRONYM.

L25: .. observed plumes: add reference

L32: (and elsewhere) : please consider to use 'trace gases' instead of 'species'

L34: Rewrite sentence

P4

L8: 'latest': A bit unclear where this refers to.

L11: 'by comparing difference MOCAGE simulations' -> 'through MOCAGE sensitivity experiments'

P2:L35, P3 L31: the year 2013 is chosen for this study, but throughout the manuscript none/or slightly varying arguments are given why 2013 is chosen. Please be consistent, and provide arguments preferably in introduction (and summary) only.

P5,

L3: summarized

L4 trajectory…'

L7: 'but does not simulate CO background': What do you mean with this phrase?

L8: '12 zones', but Figure 1 shows 14 zones?

L16: 'as the regional median': how are the regions defined here?

P6, Fig. 3: 'contribution is here mostly from NHAF'. How is 'mostly' be defined here?

L1: '2 PVU': What do you do for the tropics?

P7:

L8: 'in the CAMS project (..)'. This is not really a project. Please change to 'in the Copernicus Atmosphere Monitoring Service (CAMS)'

L10: 'six': Please check this number

L14: Could you give a few more updates on MOCAGE here? Particularly, which solver are you using, are you running in conjunction with aerosol; do you treat heterogeneous chemistry in the troposphere?

L16: '700-800 meter': Isn't this too coarse to resolve the UTLS region? Can you comment, ideally referring to potential past studies on UTLS, and sensitivity to model resolution?

L16: 'upper troposphere and **lower** stratosphere '

P8,

L10: Consider rewriting this sentence, e.g. : "MOCAGE has already been used to study various aspects of the chemical composition of the atmosphere, including the structure of the UTLS, and biomass burning plume
transport."

L16 "enhancement was due to a biomass burning plume"
L16-17 "in order to correctly represent intercontinental transport of the plume"
L17: 'in this study': Which study do the authors refer to here?

L19: '…and their direct impact to the aerosol budget over the Mediterranean basin'

L27: 'MACCity'

P9
L2: 'GFAS'. It doesn't hurt to add an extra reference here.
L4: 'emitted quantities' -> 'emission of trace gases and aerosol' ?
L7 'overall'-> multiyear
L11: repartition -> repartitioning
L13: 'The choice…' -> In this approach the injection height was set depending on …
L14: 'above the maximum of injection fraction'?

L18 'front of the fire'

P11, L21: Please explain acronym 'TTL'

P15
L5 'using the GFAS', 'carbon monoxide'. This sentence is very unclear, by the way.
L6: "The results are.. " also rewrite this sentence

L9: 'at this pressure' -> 'at this pressure level'/ at this altitude range'
L11: 'parallel' -> latitude
L12-13: confusing sentence. Please rewrite into something like 'It results in less CO being found for simulation INJ at this altitude (..) and more North ..'

P17,
L3: 'consistently enough' strange use of wording 'consistently'. Maybe change to 'with sufficient amounts' ?
L6 'performed'->'created'
L10: 2000m : what is the corresponding pressure level?
L15 'repartition'-> 'repartitioning'

P19
L8 'vertical gradient of CO in the UTLS and concentrations are captured' This is not very clear.

L12: 'IndiaS'

P21

L2 "the impact" of what?

L5 : '2 PVU'

L9 "simulation to the" : "simulation from the"

L10 "yearly"-> "annual mean"

L16 biomass burning, **namely** …

L18: "42 and 20% of CO emissions": which CO emissions are referred here? Total? Total fire ? regional?

L19 "driven" -> transported

L19  while **a fraction of** boreal fire emissions

L20 "(**approximately** 2 months)"

P23

L12  the authors write "Australian forest are also not known for their pyroconvective events".

I'm not sure about this statement. I think there have been fire events reported, even in literature, which also contain pyroconvection. Please check.

P24

L8: "This study aimed at improving the understanding **the contribution** of biomass burning emissions to upper tropospheric carbon monoxide"
L10: "We have chosen that this study would span over a single year" suggest to reformulate to something like "We have chosen a single year simulation, …"
L11: "efficiency" -> "costs"

P25

L1: 'dur'->due

L1: "… pyroconvection **for a selection of large fire events"**

L5: "however"-> "nevertheless"

L6: "It was also shown.." Please check and rewrite sentence

L7 "To go a step…" please check and rewrite sentence

L18-L19: Please check/rewrite sentence

L25-L31: Please check/rewrite sentence to be more accurate and improve readability.

---

## Referee Comment (RC2) · Anonymous Referee #2 · 5 Mar 2020

This study presents analyses of carbon monoxide (CO) distributions in the upper troposphere and lower stratosphere in connection with its sources over different regions using the global chemistry transport model, MOCAGE and also SOFT-IO, which calculates Lagrangian backward trajectory of the air parcels. The surface emissions inventory used in this study is GFAS and the model results are compared with the comprehensive airborne measurements obtained from IAGOS. My main comment is about the motivation and background of this study. Why is CO important in the upper troposphere? Has there been an issue with the injection height in global chemistry transport models in general? I believe improving the goals and motivations of this study will improve the quality of the manuscript significantly.

[Figure]

General Comments:

1. Motivation and expectation - It would be nice to see the plume injection height has been an issue in representation of CO in the global chemistry transport model, which can provide strong motivation for this work. 2. It is important to discuss the importance of UTLS CO. Why do you want to look at UTLS? Importance of UTLS CO distribution? CO in the UTLS must depend not only the emissions but also the convection in the model. 3. Results - GFAS plume rise parameters do not improve the simulation significantly. Does this mean the plume injection height is not important in general? Focus on the case where injection height makes difference instead of presenting all the cases. 4. Writing can be improved. Some of the detailed comments are provided below.

Specific Comments:

P1, L13 - This was done by comparing simulations 'were' -> Could this be 'with' instead?

P2, L12 - hydroxyl (OH) radicals -> hydroxyl radical (OH)

P2, L15 - CO can also be a way to discriminate air from the troposphere and the stratosphere, since it is only found in very low amount above the tropopause. -> This is somewhat misleading. CO decreases rapidly right above the tropopause and increases due to chemical production (For example, see Fig. 9 of Schoeberl et al., 2008JGR).

P2, L17 - transported up to the -> transported in to the

P2, L19 - thanks to deep convection -> due to deep convection

P3, L3-4 - ...the sensibility to the injection of CO from biomass burning... -> This sentence is not complete. Please consider revising.

P3, L5-7 – The reference (Deeter et al., 2013) is more appropriate for the MOPITT data. Either use MOPITT instead of IASI as an example or revise the sentence here.

[Figure]

P3, L10 – air planes –> airplanes

P3, L14-16 – I recommend revising this sentence for clarity. For instance, what does 'source appointment' mean?

P3, L17 – discriminate sources of CO anomalies encountered by the aircraft -> identify...the aircraft measurements?

P3, L22 – It would be helpful to include why considering plume injection height matters here in addition to the citation.

P4, L18 - Carbon monoxide measurements begun -> were begun

P5, Figure 1 – A description of Figure 1 should be included in the text.

P5, L3-4 – The complete method...features. -> The complete description of the method can be found in Sauvage et al. (2017). Here is the summary of its main features.

P5, L5-L9 – References for FLEXPART, ECMWF and MACCcity should be included here.

P5, L13 – attribute to

P6, Figure 3 – It should be mentioned how the CO_anomaly is calculated and what it represents here. Does it represent one plume? Why is it called anomaly?

P7, L1 – Does 'superior' mean anything larger than the anthropogenic sources even if the difference is very small?

P7, Table 1 – Are those 6 regions chosen as they have the largest numbers of plumes out of 14? It has to be mentioned in the text.

P8, L11 – What does 'impact of climate' refer to? Is this a current climate or change for the future?

P8, L20 – important fires -> fires

P8, L21 – Here a different study. . .exploring -> Our study explores

P8, L24 – taken here from – taken from

P9, L3-4 – References for GFAS and MODIS should be included.

P9, L13 – I think the injection height not only depends on the latitudes but the kinds of fires, e.g., forest fires, bush fires and etc.

P15, L5 – carbon monoxyde –> carbon monoxide

P16, Figures 10 & 11 – I don't think the differences between the Figs. 10 & 11 are significant. Either including one of them or emphasize the differences.

P20, Figure 13 – Here, results from the MOCAGE INJH runs are compared with IAGOS data. I am curious how MOCAGE BASE would look like.

P21, L6 – Around the equator -> Near the equator P21, L18-19 – and transport. . .troposphere. -> Needs a reference for this statement.

P23, L10-12 – Needs citation here.

P24, L8-10 – I would like to see the examples of contribution from the biomass burning is poorly represented in the UTLS to make this as a strong case.

Full names for all the acronyms should be provided in the manuscript. So, please double check.

---

## Author Comment (AC1) · 9 Jun 2020

We would like to thank the Anonymous Referee #1 for the detailed comments about our work, as they have been very helpful to improve our study. Our response is organised as follows. After each one of the referee's comments (in black) can be found the authors' response (in blue) followed, if needed, by changes made in the manuscript (in dark blue). In the revised version of the manuscript, only the significant changes have been coloured in blue to help identifying any new content.

This manuscript provides a nice study on the impact of biomass burning injection height parameterization on upper tropospheric CO, and the general contribution of biomass burning emissions on the upper tropospheric CO burdens, by using information on the injection height of biomass burning emissions as available from the GFAS product, the SOFT-IO tools for attributing emission sources, and the MOCAGE CTM. The study is overall clear, and well-designed. Nevertheless, I have several concerns which the authors should address before this can be accepted for publication.

1. The structure of the manuscript is not ideal. I was expecting a general model validation of (upper-) tropospheric CO near the start, but this is only covered Sec. 4.1. I suggest to shift this to Sec. 3, e.g. just before Sec. 3.1

We agree that finding a model evaluation after the beginning of the study is uncommon. Because of misleading titles in section 3, it was not clear that a first part of the model evaluation was done there, focusing on biomass burning plumes. Then, in section 4, we had presented the second part of the model evaluation using the best plume height representation on the basis of Section 3's evaluation. This is why we decided to change the titles of section 3 in the revised manuscript without changing the order of the content. To clarify the purpose of each section, we have made the following changes:

- Section 3 title is changed to: "Improvement of biomass burning injection height in MOCAGE"
- We have renamed Section 3.1 "Global evaluation of the plume representation" and Section 3.2 "Global impact of the plume representation".
- Also, subsections 4.1 and 4.2 are split into their own section; section 4 and section 5; as they are two distinct steps in our study. It also makes more sense as the general evaluation section (former Sect. 4.1) has been extended.

2. Although the evaluation metrics can well be defended (pdfs, MNMB, FGE, r), the data-aggregation is not fully suitable. There are three instances for this.

a. when evaluating the impact of injection height in Table 3, I believe the authors should equally evaluate the improvement (if any) against IAGOS measurement that do not contain biomass burning contribution. I acknowledge that this are many more instances, (as already seen from Fig. 3) which implies that the changes are expected to be small. But in any case it'd be a good to check if there is no degradation, and hopefully an improvement, in this respect.

We choose not to display statistics comparing the BASE and INJH simulations against IAGOS measurement that do not contain biomass burning contribution because we could not find any significant change. We include below the statistical indicators for both simulations, showing that these statistics are almost exactly the same. We have added in the manuscript a sentence explaining this point (P15, L1-3 in the revised version).

| BASE | | | | | | | | | | |
|---|---|---|---|---|---|---|---|---|---|---|
| | Mid to high latitudes | | | | | Tropics | | | | |
| | **2013** | DJF | MAM | JJA | SON | **2013** | DJF | MAM | JJA | SON |
| MNMB | **0.02** | 0.02 | 0.04 | 0.01 | 0.01 | **0.03** | 0.04 | 0.06 | 0.02 | 0.01 |
| FGE | **0.18** | 0.15 | 0.19 | 0.20 | 0.19 | **0.16** | 0.16 | 0.17 | 0.17 | 0.16 |
| Correlation | **0.84** | 0.85 | 0.87 | 0.82 | 0.82 | **0.75** | 0.76 | 0.73 | 0.74 | 0.76 |

| INJH | | | | | | | | | | |
|---|---|---|---|---|---|---|---|---|---|---|
| | Mid to high latitudes | | | | | Tropics | | | | |
| | **2013** | DJF | MAM | JJA | SON | **2013** | DJF | MAM | JJA | SON |
| MNMB | **0.02** | 0.02 | 0.05 | 0.01 | 0.01 | **0.03** | 0.04 | 0.06 | 0.02 | 0.01 |
| FGE | **0.18** | 0.15 | 0.19 | 0.20 | 0.19 | **0.17** | 0.16 | 0.17 | 0.17 | 0.16 |
| Correlation | **0.84** | 0.85 | 0.87 | 0.82 | 0.82 | **0.75** | 0.76 | 0.73 | 0.74 | 0.76 |

b. Likewise, in Table 4 the authors present a general overview of the statistical scores in the UTLS, in terms of MNMB, FGE and r. But as the authors discuss, there are large gradients in CO in this region, which makes it hard to judge from these statistics to what extend these are captured. Unfortunately this is also not really visible from Fig. 13, which overall focusses on the free troposphere. Therefore I suggest that the authors also present metrics specific for below and above the tropopause, using the 2 PVU metric + / - a pressure range, e.g. as the authors already specify on Page 21, line 5. Additionally a figure presenting the vertical profile of CO in the UTLS region, ideally on the vertical axis the pressure level with respect to the tropopause altitude, would be very instructive.

Indeed, our evaluation was not detailed enough to correctly evaluate MOCAGE in the UT and LS. Following the beforementioned recommendations, our study now includes:
- A new figure representing the vertical profile specifically for the UTLS region, with a vertical pressure coordinate referenced to the dynamical tropopause (with adjustments in the tropics). These profiles are computed using cruise flight measurements, in three areas of the globe defined in Fig. 1 of the manuscript, and selected for their density in IAGOS measurements in the UTLS.
- The statistical analysis in Table 4 now features separated scores for the UT and the LS, and have been split into three geographical regions instead of two (the same as for the tropopause referenced profiles). The text is also modified to add comments on this distinction in the statistical analysis.

c. Finally, for Figure 13, it'd be good to discuss (or even present) similar evaluations, but specific for the four seasons, (or at least for summer and winter seasons), considering that there is a rather large seasonal cycle in tropospheric CO, and likewise variations in general CTM abilities to capture this, which would be good to quantify.

Following your suggestion, the seasonal vertical profiles at airports are now presented in the supplementary material associated to the article. They were not originally included in the study as not all clusters are sampled enough to produce a vertical profile for each season of the year. This is why some of the figures are left blank. The conclusions from these seasonal evaluations are very similar to the one made for the yearly evaluations, with a good performance of MOCAGE through the free and upper troposphere, regardless of the season.

3. Although the manuscript is overall clearly written, there are still many issues. The introduction lacks many references, many acronyms are not defined, sentences are often not in proper English, and there are many typo's.

Following comments from both Anonymous Referees, the manuscript was improved, as some sentences were unclear, typo's remained, and acronyms were not defined everywhere in the manuscript. References were also added where missing.

4. Finally, the authors emphasize the impact of pyroconvection throughout the manuscript, but an important impact of the revised injection height parameterization appears that overall much more is injected at lower altitudes compared to the reference configuration, whereas not even a single event where a likely pyroconvection has been sampled by IAGOS, is assessed in a little more detail. This makes the reference to this process a little out of balance.

We agree that the presentation of the role of pyroconvection was not accurate and out of balance. As you spotted, the injection height is lower in boreal regions on average in INJH compared to BASE. Therefore, the improvement provided by INJH comes from the fact that it provides an actual estimate of the pyroconvection height for each individual fire while BASE gives only one height per latitude band. We have changed the manuscript in order to explain that taking the variability of plume injection height linked to pyroconvection in MOCAGE into account, improves its performances in the boreal regions. Another result on this subject, from Figs. 10, 11 and 12 (original manuscript), is that the new parameterization (INJH) induces changes compared to BASE simulation in the distribution of CO in the mid-latitudes, in the whole troposphere and lower stratosphere. This is partly related to transport processes that can occur from boreal regions to other regions (e.g. Brocchi et al. 2018). Also, the original injection parameterization leads to a sharp discontinuity at 60° latitude while in reality, the change of the height of fire injection occurs smoothly from the boreal regions to the mid-latitudes. This is another argument showing the improvement provided by the use of GFAS injection height parameters.

**Other general comments**
P2
L25: To put the emission numbers in perspective, can the authors provide an estimate of the secondary CO production from VOC and CH4 oxidation?
The following sentence was added at the end of the paragraph:
"For comparison purposes, the global in-situ productions of CO from VOCs and CH4 oxidations are respectively estimated to range between 450-1200 TgCOyr$^{-1}$ and 600-1000 TgCOyr$^{-1}$ (Stein et al., 2014)."
L29: GFAS, GFED, IS4FIRES: Missing references.
References have been added.

P3: L9: The authors quickly argue that the use of airborne campaigns is limited. This is a very short conclusion. I think this deserves a little more elaboration, e.g. including a few studies based on campaign data, and/or reporting on some interesting findings.
We agree that the comment on the field campaign was too short and limited. We have added text and references in the revised manuscript (P3, L16-20 of the revised manuscript):

P8.
L2'both anthropogenic and biogenic emissions are injected into the 5 lowest levels'. Why are biogenic emissions not injected at the surface layer?
The injection in the 5 lowest model levels is done to prevent too strong vertical gradient of emitted compounds in the PBL when emission occurs, which can cause numerical issues to the semi-lagrangian transport scheme. It is applied to all surface emissions, anthropogenic or biogenic.

L3: 'with an exponential decay': what do you mean?

The injection fraction of the mass emitted $\delta r_e$ decays with levels $L$ above the surface ($\delta r_e(L) = 0.5\delta r_e(L+1)$), ensuring a majority of emissions are still injected in the surface layer. The manuscript is modified to feature this explanation.

L10-L23: in this section I am missing a reference where the overall performance of MOCAGE tropospheric chemistry is given; now bits and pieces are given from other references. This links a bit to my request to start Sec 3.1 with a general evaluation of tropospheric CO, if there is no suitable general evaluation reference existing.

A paper on global MOCAGE performance is currently being prepared, but cannot yet be cited here. We decided to expand the current evaluation section to address this issue, following your general comments 2.a, 2.b and 2.c.

P12, L1-L4: The authors show that in simulation INJH MOCAGE captures better the observed events with high CO (particularly those > 200 ppb), which they attribute to the higher injection height for the largest fires. However, from Figure 7 it is actually clear that injection of majority of fire emissions is now actually at much lower altitudes than before, which could possibly imply that for background conditions the MOCAGE now shows a larger negative bias in the UTLS than before? Could the authors comment?

Indeed, our analysis is somewhat confusing. Our point is that despite biomass burning emission now being injected lower in the atmosphere on average, the variability of injection height is now represented. We believe that this leads to a better representation of the pathways (either direct pyroconvection or emission + advection + convection) that biomass burning emissions can take towards the UTLS. While appearing contradictory, a lower average injection height can result in stronger plumes being represented in the UTLS. The text is modified to better present our explanations for these results.

Regarding the background conditions in MOCAGE, we can see in our answer to comment 2.a that CO bias in the UTLS remains very low through the year regardless of the injection scheme.

In this line of thinking, to evaluate purely the effect of the injection height, it could be interesting to correct for the effect of a potential background model bias in CO, which also interferes with the evaluations shown in Figures 6, 8 and 9, which intend to focus on the effect of injection height, not the effect of potential model biases.

As shown in our statistical evaluation of MOCAGE, bias outside of biomass burning plumes is lower than the one seen in Fig. 6, 8 and 9 of the original manuscript. The strong negative bias seen in these figures is probably the result of the horizontal and vertical resolution of the model, resulting in artificial diffusion within the cells and preventing MOCAGE to simulate the highest values of CO mixing ratios reached in biomass burning plumes sampled at a much higher resolution by the IAGOS aircrafts.

L7: "under-represents the CO enhancement": I think in this configuration this should read 'CO amount', as the authors did not actually compute the enhancement (see previous comment) revised
L8: "diffusive": I think this is only part of the explanation. Please consider general model biases. E.g. here it would be useful to directly refer to a corresponding evaluation for against IAGOS observations where likely fire emission contributions are excluded.

This is true that general model biases can be part of the explanation. Biomass burning is an important contributor to the UTLS concentrations once plumes are diluted in the atmosphere and therefore removing the contribution of fire emissions in MOCAGE and comparing with IAGOS will not give an estimate of the model background biases but of the biomass burning contribution. However, from the new figures and the new statistical indicators provided for the UT and the LS separately, the model tends to show a slight negative bias in the UT with respect to the whole IAGOS cruise data. Thus, we

have added text on the general model bias in the UT that also explains at least part of the negative MNMB.

P15
L1-4 "to summarize…" I think these two sentences contradict each other. Either the original injection profile is good, or the GFAS product is good. Also the authors currently only focus on the impact in the UTLS. Although interesting, the largest impact of a different injection height might actually be for plumes at lower altitudes, where profiles on average look very different.
We agree that the message from these sentences was contradictory. We have changed the text to :
"Although the original representation of biomass burning injection height in MOCAGE was giving fairly good results, using GFAS improves largely the ability of MOCAGE to forecast biomass burning plumes in boreal regions. This is because GFAS provides an estimate of the actual height of pyroconvection for each fire in these regions, and therefore captures the variability of injection heights. Combined with vertical transport processes, mainly convection, it allows the model to capture well boreal plumes."
We agree that investigating implications at lower altitudes would be interesting, but the subject of the paper was the transport towards the UTLS. Also, in the study of Rémy et al (2019), where they used the same products in the C-IFS model, they found that improvements are greater for high altitude plumes (at least 4 km). This is now mentioned in the introduction of the article (P4, L14-15).

P18, Table 4: The overall small MNMB may not be very meaningful considering the large vertical gradients in this altitude range. Therefore I would appreciate if the authors could also compute the metrics for below and above the tropopause separately.
See our answer to General comment 2.b
L15. 'MOCAGE seems to correctly represent the transition..' I find this difficult to judge. Here A figure showing the vertical profiles in CO with respect to the tropopause level would be very useful.
See our answer to General comment 2.b

P19,
L18: 'realistic UTLS' -> 'realistic free tropospheric' ? revised
Figure 13: It'd be good to show this kind of evaluations for the different seasons separately. (or alternatively in the supplementary material)
See our answer to General comment 2.c

P25
L15: "as well as its gradient in the UT". First of all, the vertical profiles as presented so far were not very clear in presenting the gradient in the UT. Secondly, it is not clear what is concluded from this evaluation.
Since we now present tropopause referenced vertical profiles, we can now conclude on MOCAGE ability to represent the CO vertical gradient in the UT and LS. Sentence is changed to clarify conclusion from MOCAGE UTLS evaluation.

**Specific comments**

Abstract: Please use present or past tense consistently. Revised to present tense only.
L6: The use of these GFAS products lead**s** to improved MOCAGE skill to simulate fire plumes originating from boreal **forest** wildfires. Revised
L10: 'as the previous one': which 'one'? Changed to 'Original method of injection' for clarity
L11: database -> observations Revised
L13: were -> where; … biomass emissions were toggled … Revised

Throughout the manuscript: Please check where you use 'quantity': I think it is better to use 'amount'
Revised

P2:
L5: great -> large revised
L8: focus is made here -> here we focus revised
L11: Methane (CH4) **and VOC** … revised
L13 'carbon dioxide' revised
L14: 'due to their shorter lifetime': this is not generally true. E.g. C2H6 has a lifetime of approx. two months. revised to "as their lifetime is generally shorter"
L23: 'in the **free** troposphere' ? revised
L23: since the occurrence of this phenomenon … revised
L34: class-> classification. Also good to add reference here? revised + reference to Friedl et al. (2002)

*Friedl, M. A., McIver, D. K., Hodges, J. C. F., Zhang, X. Y., Muchoney, D., Strahler, A. H., Woodcock, C. E., Gopal, S., Schneider, A., Cooper, A., Baccini, A., Gao, F., and Schaaf, C.: Global land cover mapping from MODIS: algorithms and early results, Remote Sensing of Environment, 83, 287–302, https://doi.org/10.1016/S0034-4257(02)00078-0, 2002.*

L35: Their intensity: Unclear which intensity is meant here.
Since the point is on pyroconvection, the first part of the sentence was removed because confusing.
"Depending on the intensity of the fire as well as the atmospheric conditions above the fire, pyroconvection can occur."

P3:
L1: 'pyroconvection can occur': please add references here.
References added (Damoah et al., 2006; Cunningham and Reeder, 2009):

*Damoah, R., Spichtinger, N., Servranckx, R., Fromm, M., Eloranta, E. W., Razenkov, I. A., James, P., Shulski, M., Forster, C., and Stohl, 25 A.: A case study of pyro-convection using transport model and remote sensing data, Atmospheric Chemistry and Physics, 6, 173–185, https://doi.org/https://doi.org/10.5194/acp-6-173-2006, publisher: Copernicus GmbH, 2006.*

*Cunningham, P. and Reeder, M. J.: Severe convective storms initiated by intense wildfires: Numerical simulations of pyro-convection and pyro-tornadogenesis, Geophysical Research Letters, 36, https://doi.org/10.1029/2009GL039262, publisher: JohnWiley & Sons, Ltd, 2009.*

L3: sensibility-> sensitivity? revised
L5: The authors refer to IASI, but the reference Deeter et al. is for MOPITT. Please resolve this inconsistency revised
L6: 'lack vertical resolution in UTLS': I think this is not only in UTLS, but generally?
Revised to: 'lack vertical resolution, especially in UTLS'
L7 and L12: O3 -> O3 revised
L8: On **the** one hand… revised
L13: 'most likely source' -> 'most probable sources'? revised
L14: 'pathways' : you mean transport pathways, or chemical pathways.
Revised to transport pathways
L15/16: Suggest to change to: "This is why the analysis of large datasets require other kinds of additional information.." revised
L16: "Lagrangian backward transport calculation": add reference?
Reference added to Seibert & Franck (2004):

Seibert, P. and Frank, A.: Source-receptor matrix calculation with a Lagrangian particle dispersion model in backward mode, Atmospheric Chemistry and Physics, 4, 51–63, https://doi.org/https://doi.org/10.5194/acp-4-51-2004, publisher: Copernicus GmbH, 2004.

L20: …ability of global simulations performed by the MOCAGE … . Also please add reference to MOCAGE, and consider explaining ACRONYM. revised

L25: .. observed plumes: add reference.

We added the reference to Remy et al. (2017), already cited further in the manuscript

L32: (and elsewhere) : please consider to use 'trace gases' instead of 'species' revised

L34: Rewrite sentence. Changed to:

"In this paper  we present an implementation of the latest GFAS daily products  to constrain plume rise in the MOCAGE CTM."

P4

L8: 'latest': A bit unclear where this refers to.  Changed to 'recently released'

L11: 'by comparing difference MOCAGE simulations' -> 'through MOCAGE sensitivity experiments' revised

P2:L35, P3 L31: the year 2013 is chosen for this study, but throughout the manuscript none/or slightly varying arguments are given why 2013 is chosen. Please be consistent, and provide arguments preferably in introduction (and summary) only.

Explanations previously provided in Sect. 2.1.1 are moved in the introduction.

In the summary, the sentence is revised to be consistent with the arguments given in the introduction.

P5

L3: summarized revised

L4 trajectory…' revised

L7: 'but does not simulate CO background': What do you mean with this phrase?

SOFTIO gives an estimate of contribution of recent biomass burning emissions to CO concentrations in the UT, but not the total CO mixing ratios, which depend on more processes. Text is modified to answer this point:

"…, it results in an estimation of the contribution of recent emissions (less than 20 days) to the observed CO mixing ratios, but not of the total CO concentrations as it does not simulate CO background."

L8: '12 zones', but Figure 1 shows 14 zones? revised to 14

L16: 'as the regional median': how are the regions defined here?

The regions are defined as in Sauvage et al. (2017). Reference was added in the text.

P6

Fig. 3: 'contribution is here mostly from NHAF'. How is 'mostly' be defined here?

A new figure is presented clarifying SOFTIO contributions within the plume. The legend is also modified and now includes quantification of the anthropogenic and biomass burning total contributions.

L1: '2 PVU': What do you do for the tropics?

The following explanation is added:

In the tropics, where the definition of the potential vorticity diverges, the aircraft is always considered below the tropopause as the highest cruise altitude is below 12 km.

P7:

L8: 'in the CAMS project (..)'. This is not really a project. Please change to 'in the Copernicus Atmosphere Monitoring Service (CAMS)' revised

L10: 'six': Please check this number revised to eight

L14: Could you give a few more updates on MOCAGE here? Particularly, which solver are you using, are you running in conjunction with aerosol; do you treat heterogeneous chemistry in the troposphere?

The following explanations were added:

The solver used follows a fully implicit discretization method, described in detail in Cariolle et al. (2017). Aerosols, though described in MOCAGE (Guth et al., 2016, 2018), are not activated in this study and thus are not discussed in the present paper. Heterogenous chemistry is treated in the stratosphere but not in the troposphere.

L16: '700-800 meter': Isn't this too coarse to resolve the UTLS region? Can you comment, ideally referring to potential past studies on UTLS, and sensitivity to model resolution?

The 700-800 m vertical resolution, though not ideal, is in the medium range for UTLS simulations. The following comment is added in the manuscript:

"The vertical resolution, though not ideal, can be considered as medium and enables a representation of the main dynamical characteristics of the UTLS (Miyazaki et al., 2010; Geller et al., 2016)."

*Geller, M. A., Zhou, T., Shindell, D., Ruedy, R., Aleinov, I., Nazarenko, L., Tausnev, N. L., Kelley, M., Sun, S., Cheng, Y., Field, R. D., and Faluvegi, G.: Modeling the QBO—Improvements resulting from higher-model vertical resolution, Journal of Advances in Modeling Earth Systems, 8, 1092–1105, https://doi.org/10.1002/2016MS000699, 2016.*

*Miyazaki, K.,Watanabe, S., Kawatani, Y., Tomikawa, Y., Takahashi, M., and Sato, K.: Transport and Mixing in the Extratropical Tropopause 30 Region in a High-Vertical-Resolution GCM. Part I: Potential Vorticity and Heat Budget Analysis, Journal of the Atmospheric Sciences, 67, 1293–1314, https://doi.org/10.1175/2009JAS3221.1, 2010.*

L16: 'upper troposphere and **lower** stratosphere' revised

P8
L10: Consider rewriting this sentence, e.g. : "MOCAGE has already been used to study various aspects of the chemical composition of the atmosphere, including the structure of the UTLS, and biomass burning plume transport." revised
L16 "enhancement was due to a biomass burning plume" revised
L16-17 "in order to correctly represent intercontinental transport of the plume" revised
L17: 'in this study': Which study do the authors refer to here? Replaced 'this' by 'the previously mentioned' for clarity.
L19: '…and their direct impact to the aerosol budget over the Mediterranean basin' revised
L27: 'MACCity' revised

P9
L2: 'GFAS'. It doesn't hurt to add an extra reference here. revised
L4: 'emitted quantities' -> 'emission of trace gases and aerosol' ? revised
L7 'overall'-> multiyear revised
L11: repartition -> repartitioning revised
L13: 'The choice…' -> In this approach the injection height was set depending on … revised
L14: 'above the maximum of injection fraction'? revised
L18 'front of the fire' revised

P11,
L21: Please explain acronym 'TTL' revised to : 'tropical tropopause layer (TTL)'

P15
L5 'using the GFAS', 'carbon monoxide'. This sentence is very unclear, by the way.

The whole sentence was revised:

"In order to investigate the impact of the new injection scheme (using GFAS plume rise parameters) on carbon monoxide distribution in the upper troposphere, we perform a global comparison between the BASE and INJH MOCAGE simulations."

L6: "The results are.. " also rewrite this sentence

Sentence rewritten to:

"We treat monthly averaged results, but only August 2013 is shown hereafter as it is the month when the greatest difference between the two simulations can be seen."

L9: 'at this pressure' -> 'at this pressure level'/ at this altitude range' revised

L11: 'parallel' -> latitude revised

L12-13: confusing sentence. Please rewrite into something like 'It results in less CO being found for simulation INJ at this altitude (..) and more North ..'

The sentence was changed to:

"It results in less CO being found for simulation INJH at this altitude than in the BASE simulation (up to -25 ppb) North of the 60°N parallel, and more CO (up to +15 ppb) South."

P17

L3: 'consistently enough' strange use of wording 'consistently'. Maybe change to 'with sufficient amounts' ? revised

L6 'performed'->'created' revised

L10: 2000m : what is the corresponding pressure level?

The following was added : 2000 m (around 800 hPa)

L15 'repartition'-> 'repartitioning' revised

P19

L8 'vertical gradient of CO in the UTLS and concentrations are captured' This is not very clear.

Indeed, this part of sentence was removed, but more of the UTLS gradient is now presented with the tropopause referenced profiles (see our answer to general comment 2.b)

L12: 'IndiaS' revised

P21

L2 "the impact" of what? Of biomass burning emission, revised

L5 : '2 PVU' revised

L9 "simulation to the" : "simulation from the" revised

L10 "yearly"-> "annual mean" revised

L16 biomass burning, **namely** ... revised

L18: "42 and 20% of CO emissions": which CO emissions are referred here? Total? Total fire ? regional?

Revised to: '42 and 20% of CO emissions from biomass burning'

L19 "driven" -> transported revised

L19 while **a fraction of** boreal fire emissions revised

L20 "(**approximately** 2 months)" revised

P23

L12 the authors write "Australian forest are also not known for their pyroconvective events". I'm not sure about this statement. I think there have been fire events reported, even in literature, which also contain pyroconvection. Please check.

Indeed, they are example of pyroconvective events in Australia (Dowdy et al., 2017, 2018), and the point should have been kept to the strong influence of deep convective transport above the Maritime Continent. This statement has been removed from the manuscript, and following comments from Anonymous Referee #2, references were added on pathways of biomass burning emissions to the upper troposphere (see AC2).

*Dowdy, A. J., & Pepler, A. (2018). Pyroconvection risk in Australia: Climatological changes in atmospheric stability and surface fire weather conditions. Geophysical Research Letters, 45, 2005– 2013. https://doi.org/10.1002/2017GL076654*

*Dowdy, A. J., Fromm, M. D., and McCarthy, N. (2017), Pyrocumulonimbus lightning and fire ignition on Black Saturday in southeast Australia, J. Geophys. Res. Atmos., 122, 7342– 7354, doi:10.1002/2017JD026577.*

P24
L8: "This study aimed at improving the understanding **the contribution** of biomass burning emissions to upper tropospheric carbon monoxide" revised
L10: "We have chosen that this study would span over a single year" suggest to reformulate to something like "We have chosen a single year simulation, …" revised
L11: "efficiency" -> "costs" revised

P25
L1: 'dur'->due revised
L1: "… pyroconvection **for a selection of large fire events"** revised
L5: "however"-> "nevertheless" revised
L6: "It was also shown.." Please check and rewrite sentence revised
L7 "To go a step…" please check and rewrite sentence revised
L18-L19: Please check/rewrite sentence revised
L25-L31: Please check/rewrite sentence to be more accurate and improve readability. revised

---

## Author Comment (AC2) · 9 Jun 2020

We would like to thank the Anonymous Referee #2 for the detailed comments about our work, as they have been very helpful to improve our study. Our response is organised as follows. After each one of the referee's comments (in black) can be found the authors' response (in blue) followed, if needed, by changes made in the manuscript (in dark blue). In the revised version of the manuscript, only the significant changes have been coloured in blue to help identifying any new content. Please note that the structure of the manuscript has been slightly modified to clarify each sections purpose in the paper (see our answer to Anonymous Referee #2 for more details).

This study presents analyses of carbon monoxide (CO) distributions in the upper troposphere and lower stratosphere in connection with its sources over different regions using the global chemistry transport model, MOCAGE and also SOFT-IO, which calculates Lagrangian backward trajectory of the air parcels. The surface emissions inventory used in this study is GFAS and the model results are compared with the comprehensive airborne measurements obtained from IAGOS. My main comment is about the motivation and background of this study. Why is CO important in the upper troposphere? Has there been an issue with the injection height in global chemistry transport models in general? I believe improving the goals and motivations of this study will improve the quality of the manuscript significantly.

General Comments:
1. Motivation and expectation - It would be nice to see the plume injection height has been an issue in representation of CO in the global chemistry transport model, which can provide strong motivation for this work.
Indeed, the justification to the choice of the injection height implementation was not developed enough to justify our motivations for this work. In addition to our response to one of the following comments (P3, L22), we added the following explanations in the introduction of the revised manuscript:
"Moreover, Fromm et al. (2019) recently carried out a reinterpretation of existing literature on the pathway of wildfires emissions to the UTLS, stating that on multiple occasion studies have wrongly attributed plumes observed in the upper troposphere to transport from traditional cumulonimbus (Cb) instead of pyrocumulonimbus (pyCb). They concluded that the phenomenon of pyroconvection has probably been overlooked and its impact underestimated in past studies, encouraging the use of reliable information on its occurrence to accurately quantify vertical transport of emissions."

2. It is important to discuss the importance of UTLS CO. Why do you want to look at UTLS? Importance of UTLS CO distribution? CO in the UTLS must depend not only the emissions but also the convection in the model.
We decided to add more comments about the importance of UTLS CO, its distribution and impacts on the overall chemical composition off the atmosphere. We also provided explanations on the role of biomass burning in these processes. Additions are made in the manuscript at two places:
- P2, L15-16 of the revised manuscript:
"Moreover, because ozone radiative impact relies mostly on its distribution in the UTLS (Riese et al., 2012), it makes CO indirectly influencing the global radiative budget of the Earth."
- P2, L31 – P3, L2 of the revised manuscript:
"As biomass burning emissions can reach rapidly the upper troposphere in the form of plume transported though convection or pyroconvection, they have been studied for their potential to contribute to ozone production at this altitude. Enhancements of ozone amounts have been observed and modelled in biomass burning plumes (Thomas et al., 2013), with production increasing while the plume ages. It can lead to export of ozone as the plume are transported by the general circulation on a hemispheric scale (Brocchi et al., 2017)."

3. Results - GFAS plume rise parameters do not improve the simulation significantly. Does this mean the plume injection height is not important in general? Focus on the case where injection height makes difference instead of presenting all the cases.

The use of GFAS injection parameters instead of 3 fixed heights depending on latitudes in MOCAGE improves significantly the forecast of fire plumes in the UT in boreal regions with respect to IAGOS cruise data. This is mainly because, there, plumes are more driven by pyroconvection than at other latitudes. Therefore, this is where we expected and we do get most improvement from the GFAS injection parameters that provide an estimate of the actual height reached by pyroconvection for each fire. Taking the variability of heights of pyroconvection into account in the model is an important outcome of this study even though there is no significant improvement on average in the UT at other latitudes with respect to IAGOS data. Another important point, from Figs. 10, 11 and 12 from the original manuscript, one can see that the new parameterization induces changes in the distribution of CO in the mid-latitudes, in the whole troposphere and lower stratosphere. This is partly related to transport processes from boreal regions to other regions (e.g. Brocchi et al. 2018). Also, the original injection parameterization leads to a sharp discontinuity at 60° latitude while in reality, the change of the height of fire injection occurs smoothly from the boreal regions to the mid-latitudes. This is another argument for the use of GFAS parameters.

All these arguments have been made clearer in the revised manuscript and the part devoted to the other regions than boreal has been reduced (fig. 9 removed).

4. Writing can be improved. Some of the detailed comments are provided below.

Following comments from both Anonymous Referees, the manuscript was improved, as some sentences were unclear, typos remained, and acronyms were not always defined in the manuscript.

Specific Comments:

P1, L13 - This was done by comparing simulations 'were' -> Could this be 'with' instead? Revised

P2, L12 - hydroxyl (OH) radicals -> hydroxyl radical (OH) revised

P2, L15 - CO can also be a way to discriminate air from the troposphere and the stratosphere, since it is only found in very low amount above the tropopause. -> This is somewhat misleading. CO decreases rapidly right above the tropopause and increases due to chemical production (For example, see Fig. 9 of Schoeberl et al., 2008JGR).
Indeed, for clarity it has been rephrased to:
"CO can also be a way to discriminate  upper tropospheric from lower stratospheric air masses, since it is only found in very low amount directly above the tropopause."

P2, L17 - transported up to the -> transported in to the revised

P2, L19 - thanks to deep convection -> due to deep convection revised

P3, L3-4 - . . .the sensibility to the injection of CO from biomass burning. . . -> This sentence is not complete. Please consider revising.
Sentence was revised as follows:
'In this study , we investigate biomass burning emissions and their impacts on CO distribution in the upper troposphere, through global modelling and in-situ measurements.'

P3, L5-7 – The reference (Deeter et al., 2013) is more appropriate for the MOPITT data. Either use MOPITT instead of IASI as an example or revise the sentence here.

Replaced 'IASI' by 'MOPPIT' in the sentence.

P3, L10 – air planes –> airplanes revised

P3, L14-16 – I recommend revising this sentence for clarity. For instance, what does 'source appointment' mean? The actual word is apportionment and the sentence has been revised accordingly.

P3, L17 – discriminate sources of CO anomalies encountered by the aircraft ->
identify. . .the aircraft measurements?
Sentence was revised as follows:
"Lagrangian backward transport calculation. SOFT-IO (Sauvage et al., 2017) is a recently developed tool coupling backward transport calculation and emission inventories to  to estimate the contribution of recent emissions to CO anomalies identified in the aircraft measurements."

P3, L22 – It would be helpful to include why considering plume injection height matters here in addition to the citation.
The following sentence was added:
"It was found that not only the vertical distribution of emitted trace gases was impacted, but also their long range transport as injection can occur directly in the upper troposphere."

P4, L18 - Carbon monoxide measurements begun -> were begun revised

P5, Figure 1 – A description of Figure 1 should be included in the text.
Figure 1 has been simplified and now only displays trajectory regardless of the IAGOS package as we focus in this study on CO which is always measured by IAGOS packages. A description of its main features is now discussed in the text:
"It is noticeable that the northern mid-latitudes are the most sampled, with two main axes for IAGOS flights: from Europe to North America going over the Atlantic Ocean, and from Europe to Eastern Asia going over Boreal Asia. The tropics flights tracks mainly cover the African continent and the maritime continent."

P5, L3-4 – The complete method. . .features. -> The complete description of the method can be found in Sauvage et al. (2017). Here is the summary of its main features. Revised

P5, L5-L9 – References for FLEXPART, ECMWF and MACCcity should be included here. Reference added.

P5, L13 – attribute to revised

P6, Figure 3 – It should be mentioned how the CO_anomaly is calculated and what it represents here. Does it represent one plume? Why is it called anomaly?
Figure and caption have been redone to be consistent with CO anomaly calculation from Eq. 1. The single plume is also visually highlighted.

P7, L1 – Does 'superior' mean anything larger than the anthropogenic sources even if the difference is very small?
Yes, it means larger. A plume is considered originating from biomass burning, even if it leads to selecting plume with mixed (anthropogenic/biomass burning) origins, we know that most of it is due to biomass burning. A sentence is added in the manuscript for clarification:

"For each CO anomaly detected during a flight, if the biomass burning contribution from SOFT-IO is on average higher than 5 ppb and is greater than the anthropogenic contribution from SOFT-IO, the anomaly is selected as a biomass burning plume. This ensures that biomass burning provides a significant anomaly with respect to CO background and that this is the main contributor."

P7, Table 1 – Are those 6 regions chosen as they have the largest numbers of plumes out of 14? It has to be mentioned in the text.
No plume has been detected from a region other than the 6 detailed in Table 1, but 29 of them come from multiple areas. The text and table description have been clarified:

"A summary of the number of plumes by origin can be found in Table 1. The choice was made to merge together plumes origination from NHAF and SHAF, as well as plumes from BONA and BOAS, as their characteristics are expected to be similar."

"**Table 1.** Number of biomass burning plumes sampled by IAGOS aircrafts in 2013, following SOFT-IO contribution calculations per geographical origin. Regions from which no plume were sampled are not shown. The MULTIPLE origin corresponds to plume having more than one possible region of origin (other than AFR and BOREAL)."

P8, L11 – What does 'impact of climate' refer to? Is this a current climate or change for the future? Changed to 'impact of present and future climate'

P8, L20 – important fires -> fires revised

P8, L21 – Here a different study. . .exploring -> Our study explores revised

P8, L24 – taken here from – taken from revised

P9, L3-4 – References for GFAS and MODIS should be included. References included.

P9, L13 – I think the injection height not only depends on the latitudes but the kinds of fires, e.g., forest fires, bush fires and etc.
Yes, the injection height depends heavily on the nature of the fuel as well as on the kind of fire (from crop, peat, …) and the meteorological environment where the fire develops, but since that information was unavailable, latitude dependent profiles were chosen. The manuscript was modified accordingly:
" In this approach the injection height was set depending on the latitude of the fire, even though it relies on other parameters like the type of fire."

P15, L5 – carbon monoxyde –> carbon monoxide revised

P16, Figures 10 & 11 – I don't think the differences between the Figs. 10 & 11 are significant. Either including one of them or emphasize the differences.
Since comments on the vertical distribution of CO can be made using Fig. 12, Fig. 11 has been removed from the manuscript as it does not provide additional information. Associated comments in the manuscript are modified and moved in the paragraph about the original Fig 12.

P20, Figure 13 – Here, results from the MOCAGE INJH runs are compared with IAGOS data. I am curious how MOCAGE BASE would look like.
MOCAGE BASE are almost identical to MOCAGE INJH results when plotted as vertical mean profiles and compared to IAGOS data, and thus have not been shown in this figure. The lack of difference comes from the fact that this figure makes use of all IAGOS data (not only biomass burning plumes)

and that there is no airport in the boreal region, only region where significant changes are expected from figure 8.

P21, L6 – Around the equator -> Near the equator revised

P21, L18-19 – and transport. . .troposphere. -> Needs a reference for this statement.
Reference added to Huang et al. (2012,2014) and Liu et al. (2013) :

> *Huang, L., Fu, R., Jiang, J. H., Wright, J. S., and Luo, M.: Geographic and seasonal distributions of CO transport pathways and their roles in determining CO centers in the upper troposphere, Atmospheric Chemistry and Physics, 12, 4683–4698, https://doi.org/10.5194/acp-12-4683-2012, 2012.*

> *Huang, L., Fu, R., and Jiang, J. H.: Impacts of fire emissions and transport pathways on the interannual variation of CO in the tropical upper5 troposphere, Atmospheric Chemistry and Physics, 14, 4087–4099, https://doi.org/10.5194/acp-14-4087-2014, 2014.*

> *Liu, J., Logan, J. A., Murray, L. T., Pumphrey, H. C., Schwartz, M. J., and Megretskaia, I. A.: Transport analysis and source attribution of seasonal and interannual variability of CO in the tropical upper troposphere and lower stratosphere, Atmospheric Chemistry and Physics, 13, 129–146, https://doi.org/10.5194/acp-13-129-2013, 2013.*

P23, L10-12 – Needs citation here.
According to comments from Anonymous Referee #1, this sentence is actually inaccurate since cases of pyroconvection have been reported in Australia, and the point was more about particular strength of convective transport over the maritime continent. Manuscript was modified accordingly.

P24, L8-10 – I would like to see the examples of contribution from the biomass burning is poorly represented in the UTLS to make this as a strong case.
We are not understanding this comment. Is this comment for P25, L8-10, instead of page 24 ? if so, here is our answer :

These two lines may have gone further than what can be concluded from our work. We decided to rewrite them to be in line with the content of the manuscript:
"Based on the results of this study and previous ones that have shown that transport through pyroconvection has been underestimated (Fromm et al.,2019), it appears that the use of products such as GFAS injection height can offer improvement in the representation of biomass burning plumes in atmospheric models, especially for mid to high latitudes."

Full names for all the acronyms should be provided in the manuscript. So, please double check. revised